# A network of heterochronic genes including *Imp1* regulates temporal changes in stem cell properties

**Jinsuke Nishino[1,2], Sunjung Kim[3,4,5], Yuan Zhu[3,4,5], Hao Zhu[1,2], Sean J Morrison[1,2,6]***

[1]Children's Research Institute, University of Texas Southwestern Medical Center, Dallas, United States; [2]Department of Pediatrics, University of Texas Southwestern Medical Center, Dallas, United States; [3]Division of Molecular Medicine and Genetics, University of Michigan, Ann Arbor, United States; [4]Department of Internal Medicine, University of Michigan, Ann Arbor, United States; [5]Department of Cell and Developmental Biology, University of Michigan, Ann Arbor, United States; [6]Howard Hughes Medical Institute, University of Texas Southwestern Medical Center, Dallas, United States

**Abstract** Stem cell properties change over time to match the changing growth and regeneration demands of tissues. We showed previously that adult forebrain stem cell function declines during aging because of increased expression of *let-7* microRNAs, evolutionarily conserved heterochronic genes that reduce HMGA2 expression. Here we asked whether *let-7* targets also regulate changes between fetal and adult stem cells. We found a second *let-7* target, the RNA binding protein IMP1, that is expressed by fetal, but not adult, neural stem cells. IMP1 expression was promoted by Wnt signaling and Lin28a expression and opposed by *let-7* microRNAs. *Imp1*-deficient neural stem cells were prematurely depleted in the dorsal telencephalon due to accelerated differentiation, impairing pallial expansion. IMP1 post-transcriptionally inhibited the expression of differentiation-associated genes while promoting the expression of self-renewal genes, including *Hmga2*. A network of heterochronic gene products including Lin28a, *let-7*, IMP1, and HMGA2 thus regulates temporal changes in stem cell properties.

*For correspondence: sean. morrison@utsouthwestern.edu

## Introduction

Stem cell properties change throughout life in many tissues in response to changing growth and regeneration demands (*He et al., 2009*). These changes are particularly evident in the central nervous system (CNS) forebrain, where neural stem cells persist throughout life. During fetal development rapidly dividing neural stem cells expand in number before differentiating in precisely defined temporal windows, first to form neurons and then to form glia (*Salomoni and Calegari, 2010*). Largely quiescent neural stem cells persist into adulthood in the lateral wall of the lateral ventricle subventricular zone (SVZ) as well as in the dentate gyrus, where they give rise to new interneurons throughout adult life (*Alvarez-Buylla and Lim, 2004*; *Zhao et al., 2008*). However, the rate of neurogenesis, the frequency of stem cells, and their rate of proliferation all decline with age (*Kuhn et al., 1996*; *Enwere et al., 2004*; *Maslov et al., 2004*; *Molofsky et al., 2006*; *Bonaguidi et al., 2011*; *Encinas et al., 2011*). A fundamental question concerns the mechanisms that control these temporal changes in stem cell properties.

The declines in SVZ proliferation, stem cell self-renewal potential, and neurogenesis during aging are regulated by a pathway that includes *let-7* microRNAs, the chromatin-associated HMGA2 high mobility group protein, and the p16[Ink4a] cyclin-dependent kinase inhibitor: *let-7b* expression increases

**eLife digest** Stem cells are found throughout the body, and play key roles in promoting tissue growth during fetal development, and in maintaining tissues in the adult. When stem cells divide, they can either give rise to more stem cells, or they can generate specialized cells required for tissue function. However, the properties of stem cells must change over time to match the changing growth and regeneration demands of tissues.

A previous study by Nishino et al. has shown that expression of a micro RNA molecule called *let-7* increases throughout adulthood, and this reduces the activity of stem cells in older animals. Now, Nishino et al. report that *let-7*, and other genes it regulates, also control the dramatic changes that occur in the properties of stem cells between fetal development and adulthood. Whereas stem cells in the fetal forebrain undergo rapid division and are capable of generating many different cell types, stem cells in the adult forebrain divide less often and can generate only a few specific types of cell. While Nishino et al. performed their study on stem cells in the brain, their results are likely to apply also to stem cells in other tissues.

Nishino et al. show that *let-7* regulates the production of an RNA binding protein called IMP1. Mice with stem cells that lack IMP1 have a smaller cerebral cortex than normal mice because their stem cells undergo fewer rounds of division before committing to become brain cells. Additional experiments revealed that IMP1 inhibits the expression of genes that trigger stem cells to commit to specific fates and promotes the expression of genes related to self-renewal.

These results indicate that the gene that encodes IMP1 is expressed in fetal neural stem cells, but not in adult neural stem cells, and that the reduced production of this protein contributes to the developmental switch from highly proliferative neural stem cells in the fetus to the more quiescent stem cells found in adults. Further studies are likely to identify many more targets of *let-7* that enable stem cells to adapt their properties to the changing needs of the organism over time.

These results are interesting because let-7-regulated networks were first discovered based on their ability to regulate the timing of developmental transitions in worms. This suggests that the mechanisms employed by mammalian tissue stem cells to regulate changes in their properties over time, are at least partly evolutionarily conserved mechanisms inherited from invertebrates.

with age, reducing Hmga2 expression and increasing $p16^{Ink4a}$ expression (**Nishino et al., 2008**). $p16^{Ink4a}$ deficiency or overexpression of a *let-7* insensitive form of *Hmga2* partially rescues the declines in neural stem cell function and neurogenesis in aging mice (**Molofsky et al., 2006**; **Nishino et al., 2008**). This pathway appears to be conserved among multiple mammalian tissues as $p16^{Ink4a}$ deficiency also increases the function of hematopoietic stem cells and pancreatic beta cells during aging (**Janzen et al., 2006**; **Krishnamurthy et al., 2006**). HMGA2 also promotes hematopoietic stem cell self-renewal (**Cavazzana-Calvo et al., 2010**; **Ikeda et al., 2011**) and myoblast proliferation (**Li et al., 2012**).

*let-7* microRNAs are evolutionarily conserved heterochronic genes that regulate developmental timing (**Pasquinelli et al., 2000**) and aging (**Shen et al., 2012**) in *Caenorhabditis elegans*. In mammals, *let-7* microRNAs are known to regulate embryonic stem cells (**Melton et al., 2010**), primordial germ cells (**West et al., 2009**), and adult neural stem cells (**Zhao et al., 2010**) but it is unclear to what extent *let-7* targets regulate developmental changes in mammalian stem cell function over time. For example, it is unclear whether the *let-7*-regulated pathway we identified in aging stem cells only regulates stem cell aging or whether it is one branch of a larger network of heterochronic genes that regulates temporal changes in stem cell function throughout life.

*let-7* microRNAs negatively regulate the expression of a number of gene products, including Insulin-like growth factor two mRNA binding protein 1 (IMP1; also known as CRD-BP and VICKZ1) (**Boyerinas et al., 2008**). IMP1 binds to target RNAs, post-transcriptionally regulating their localization, turnover, and translation (**Doyle et al., 1998**; **Nielsen et al., 1999**; **Farina et al., 2003**; **Atlas et al., 2004**). *Imp1* expression is widespread in fetal tissues but declines perinatally and is not detected in most adult tissues (**Hansen et al., 2004**; **Hammer et al., 2005**). *Imp1* expression is elevated in several cancers (**Ioannidis et al., 2004**; **Yisraeli, 2005**), partly as a consequence of Wnt signaling, which promotes *Imp1* transcription (**Noubissi et al., 2006**; **Gu et al., 2008**). Over-expression of IMP1 can promote

tumorigenesis (*Tessier et al., 2004*). *Imp1* deficient mice have a dwarf phenotype with some neonatal mortality (*Hansen et al., 2004*). However, it is unknown if IMP1 regulates stem cells.

Canonical Wnt signaling promotes a rapid expansion in the number of undifferentiated stem cells during forebrain development (*McMahon et al., 1992*; *McMahon and Bradley, 1990*; *Ikeya et al., 1997*; *Dickinson et al., 1994*; *Wrobel et al., 2007*). Wnt signaling prevents cell cycle exit and delays differentiation in these cells (*Megason and McMahon, 2002*; *Chenn and Walsh, 2002*; *Machon et al., 2003*; *Zechner et al., 2003*; *Zhou et al., 2006*; *Woodhead et al., 2006*; *Gulacsi and Anderson, 2008*; *Wrobel et al., 2007*). Wnt signaling also promotes the maintenance of stem cells in the adult forebrain (*Kuwabara et al., 2009*; *Qu et al., 2010*). However, it is unclear why Wnt signaling expands the number of neural stem cells during development but only maintains declining numbers of stem cells during adulthood.

Here we report that *Imp1* is expressed in fetal neural stem/progenitor cells as a consequence of Wnt signaling but that its expression declines in late fetal development, partly as a consequence of increasing *let-7* microRNA expression. *Imp1* promoted the expansion of fetal neural stem cells and *Imp1* deficiency reduced brain mass. IMP1 bound to a number of mRNAs, post-transcriptionally promoting the expression of gene products that promote self-renewal, including *Hmga2*, and inhibiting the expression of gene products involved in differentiation. Our findings demonstrate a novel role for IMP1 in the expansion of fetal neural stem cells and suggest that the perinatal loss of IMP1 expression is part of the mechanism that allows Wnt signaling to promote the expansion of fetal stem cells. More broadly, our results demonstrate that a network of heterochronic genes regulates temporal changes in stem cell function throughout life.

## Results

### *Imp1* is expressed by stem/progenitor cells in the pallial region of the telencephalon

We examined *Imp1* expression by quantitative RT-PCR (qPCR) in CNS stem/progenitor cells from the embryonic day (E)12.5 dorsal telencephalon, E14.5 dorsal telencephalon, postnatal day (P)0 lateral ventricle ventricular zone (VZ), and P30 lateral ventricle subventricular zone (SVZ). *Imp1* expression was high at E12.5 but declined over 100-fold by P0 and was no longer detected in the P30 SVZ (*Figure 1A*). The family members *Imp2* and *Imp3* were expressed in patterns very similar to *Imp1*: high in the telencephalon VZ at E12.5 but declining sharply throughout fetal development (*Figure 1—figure supplement 1H,I*). This raises the possibility of redundancy among IMP family members during CNS development. In contrast to the *Imp1* expression pattern, *let-7b* expression was very low at E12.5 but increased approximatly 40-fold by P0 and continued to increase into adulthood (*Figure 1B*). To confirm that *let-7b* can regulate *Imp1*, we overexpressed *let-7b* in neural stem/progenitor cells cultured from E14.5 dorsal telencephalon. IMP1 protein levels were reduced in neurospheres that overexpressed *let-7b* (*Figure 1C*). This suggests that *Imp1* expression declines as *let-7b* expression increases during fetal development and that *let-7b* can inhibit *Imp1* expression.

To systematically examine *Imp1* expression we analysed a gene-trap mouse (*Imp1^{β-geo/+}*) in which *β-galactosidase-neomycin* (*β-geo*) was inserted into the second intron of *Imp1* (*Hansen et al., 2004*). This led to the expression of an IMP1-β-geo fusion protein that contained the IMP1 RNA recognition motif encoded by exons 1 and 2 but lacked the second RNA recognition motif and the four hnRNP K homology domains, which are essential for biological activity (*Nielsen et al., 2002*). This mouse therefore provided a loss of function allele that allowed us to monitor *Imp1* expression by β-galactosidase activity (*Hansen et al., 2004*).

At E10.5, *Imp1* was expressed throughout the VZ of the entire developing brain with the exception of the floor plate and roof plate (*Figure 1—figure supplement 1A*). The anatomy of the developing forebrain and the *Imp1* expression pattern are schematically summarized in *Figure 1—figure supplement 2*. At later stages (E12.5-E16.5) *Imp1* expression was gradually restricted, mainly to the dorsomedial telencephalon (DMT), where it continued to be expressed by Pax6+ undifferentiated neural stem/progenitor cells in the VZ/SVZ (*Figure 1D,E*, *Figure1—figure supplement 1B*). There was little or no *Imp1* expression by the differentiated neurons that accumulated at the cortical plate (*Figure 1E*). At birth, there was little or no *Imp1* expression in the cerebral cortex (*Figure 1D*). In situ hybridization to endogenous *Imp1* transcripts revealed a similar expression pattern as observed with X-gal staining of *Imp1^{β-geo/+}* mice: *Imp1* was mainly expressed in the VZ/SVZ of the dorsal telencephalon

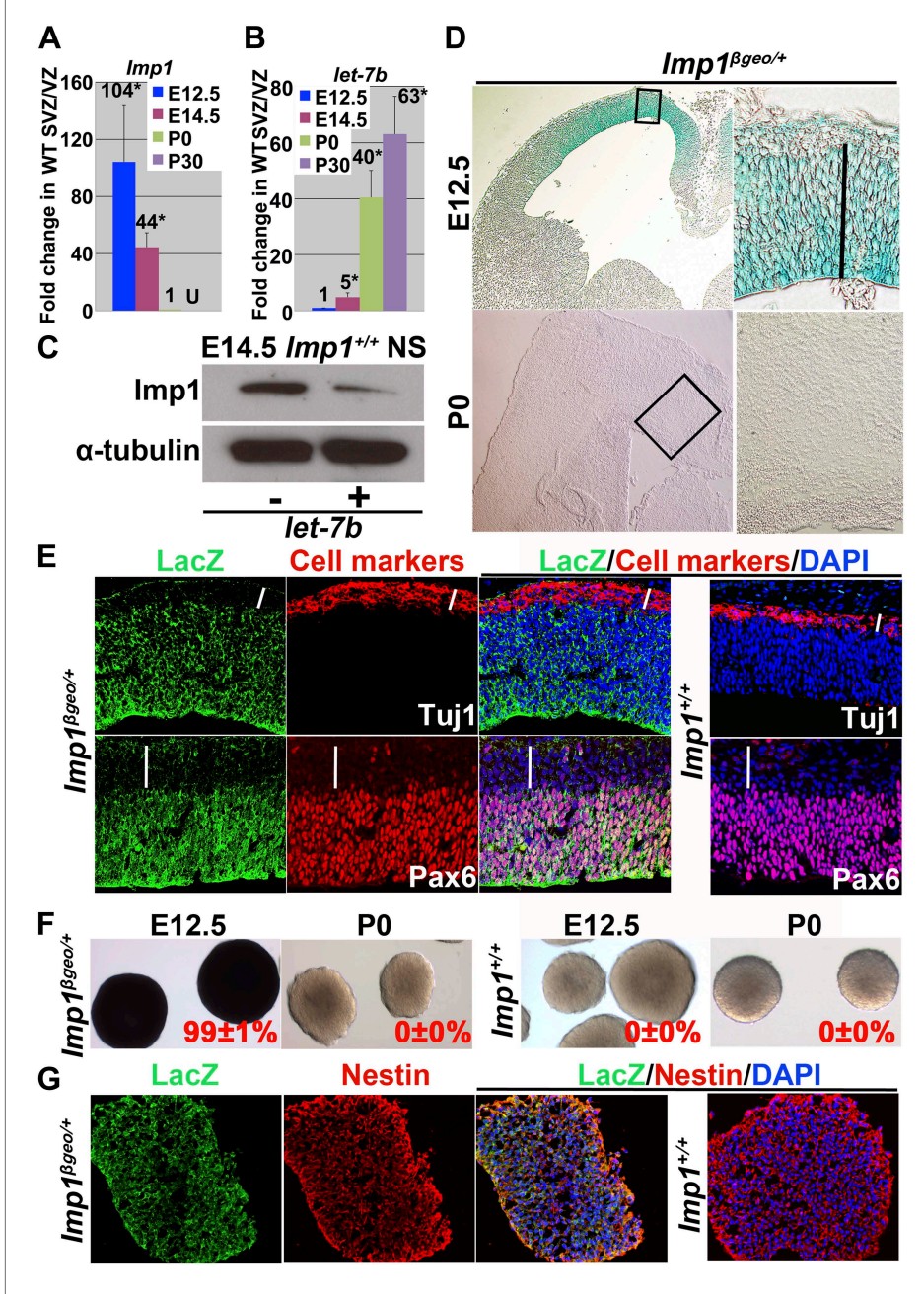

**Figure 1**. *Imp1* expression declines over time in neural stem/progenitor cells in the dorsal telencephalon and is extinguished postnatally. (**A** and **B**) qPCR for *Imp1* (**A**) and *let-7b* (**B**) in E12.5 dorsal telencephalon, E14.5 dorsal telencephalon, P0 lateral ventricle VZ/SVZ, and P30 lateral ventricle VZ/SVZ (fold change mean±SD for 3–4 mice/stage; U, not detectable above background; *p<0.01). (**C**) Western blot of E14.5 wild-type neurospheres infected with either *GFP-only* control lentivirus (−) or with *let-7b+GFP* lentivirus (+). *Let-7b* overexpression reduced IMP1 expression. (**D**) X-gal staining of sections from E12.5 and P0 *Imp1^β-geo/+* forebrain. *Imp1* was expressed in a medial-high/lateral-low gradient in the E12.5 dorsal telencephalon and confined to undifferentiated cells in the VZ/SVZ (solid line). At P0, no X-gal staining was detectable. A high magnification image is shown for the boxed area on the low magnification image to the left. See also *Figure 1—figure supplement 1A,B* for X-gal staining in E10.5, E14.5, and E16.5 brains. (**E**) Immunostaining for LacZ in the dorsal telencephalon from E13.5 *Imp1^β-geo/+* and *Imp1^+/+* mice. *Imp1* was expressed by the Pax6+ neural progenitors in the VZ but not by the TuJ1+ neurons at the cortical plate (white bars). Nuclei were visualized using 4'6-diamino-2-phenylindole dihydrocloride (DAPI) staining. (**F**) Virtually all neurospheres cultured from E12.5 *Imp1^β-geo/+* dorsal telencephalon, but not from P0 *Imp1^β-geo/+*

*Figure 1. Continued on next page*

*Figure 1. Continued*

neocortical VZ, stained with X-gal (mean ± SD % X-gal+, three experiments). (**G**) LacZ and Nestin immunostaining overlapped in sections through neurospheres cultured from *Imp1$^{β-geo/+}$* E12.5 dorsal telencephalon.

The following figure supplements are available for figure 1:

**Figure supplement 1**. *Imp1* expression in the fetal brain is spatially restricted over time and extinguished in adult brain.

**Figure supplement 2**. Schematic showing fetal telencephalon development and *Imp1* expression.

at E14.5 and was not detected in P60 hippocampus or olfactory bulb (***Figure 1—figure supplement 1E–G***).

We cultured cells from E12.5 or P0 cerebral cortex or P60 lateral ventricle SVZ in non-adherent cultures at clonal density. Almost all neurospheres formed by E12.5 *Imp1$^{β-geo/+}$* telencephalon cells, but not littermate control cells, stained with X-gal and this staining was maintained upon passaging of neurospheres (***Figure 1F***, ***Figure 1—figure supplement 1C***). Immunostaining for β-galactosidase and Nestin co-localized in sections from E12.5 *Imp1$^{β-geo/+}$* neurospheres (***Figure 1G***). We could not detect X-gal staining in neurospheres cultured from P0 or P60 *Imp1$^{β-geo/+}$* lateral ventricle VZ cells (***Figure 1F***, ***Figure 1—figure supplement 1D***). *Imp1* is therefore expressed in neural stem/progenitor cells in the fetal telencephalon but it's expression is extinguished postnatally.

## *Imp1* deficiency depleted neural stem cells, reducing pallial expansion and brain size

Consistent with an earlier report (***Hansen et al., 2004***), *Imp1* deficiency led to growth retardation in mice that was evident by late fetal development and persisted into adulthood (***Figure 2—figure supplement 1A,C***). The brains of *Imp1$^{β-geo/β-geo}$* mice were also significantly (p<0.01) smaller than the brains of littermate controls at P0 and P30 (***Figure 2—figure supplement 1B,D***). Histological analysis of E18.5 *Imp1$^{β-geo/β-geo}$* and littermate control brains showed that pallial expansion was impaired in *Imp1$^{β-geo/β-geo}$* brains (***Figure 2A***). The pial surface from the pallial/subpallial boundary to the retrosplenial cortex, was significantly (p<0.05) shortened in *Imp1$^{β-geo/β-geo}$* brains as compared to littermate controls (L in ***Figure 2A,D***). The lateral ventricle was collapsed in *Imp1$^{β-geo/β-geo}$* brains, in contrast to littermate controls (***Figure 2A***). Cortical thickness was not significantly affected by *Imp1* deficiency (T in ***Figure 2A,C***). These morphological abnormalities in the cortex first became apparent around E14.5, with a shortened pial surface length, and became increasingly severe throughout the rest of development (***Figure 2A,D***). At E16.5 and E18.5, the pial surface length was significantly shorter in the *Imp1$^{β-geo/β-geo}$* forebrain (***Figure 2D***) and the lateral ventricle collapsed (***Figure 2A***).

It is not clear whether the overall growth retardation (***Figure 2—figure supplement 1A–D***) reflects cell autonomous effects of *Imp1* deficiency in stem cells from multiple tissues or whether it reflects non-cell autonomous effects. However, the impaired pallial expansion in the dorsal cortex (***Figure 2A***) coincided precisely with the domain of *Imp1* expression that persisted throughout fetal development (***Figure 1D,E***; ***Figure 1—figure supplement 1B***, ***Figure 1—figure supplement 2***). We therefore hypothesized that *Imp1* is required cell autonomously within stem cells in the dorsal telencephalon to promote pallial expansion, a key event in cortical development.

We examined cells within the dorsal telencephalon in more detail to better understand why pallial expansion was impaired in the absence of *Imp1*. The number of proliferating cells that incorporated a pulse of BrdU in the VZ did not significantly differ between *Imp1$^{β-geo/β-geo}$* and control sections at E12.5 or E14.5 (***Figure 2B,E***). However, we detected significantly (p<0.05) lower numbers of BrdU+ cells in the VZ of the *Imp1$^{β-geo/β-geo}$* dorsomedial telencephalon at E16.5 and E18.5, and in the VZ of *Imp1$^{β-geo/β-geo}$* dorsolateral telencephalon at E18.5 (***Figure 2B,E***). The reduction in proliferating VZ cells was more prominent in dorsomedial telencephalon, where *Imp1* is normally strongly expressed, as compared to the dorsolateral telencephalon where *Imp1* is more weakly expressed (***Figure 2E***).

We did not observe any of these effects in *Imp1$^{β-geo/+}$* heterozygous mice (***Figure 2E***), demonstrating that these effects reflect a loss of IMP1 function rather than a gain-of-function associated with the mutant allele. Moreover, retroviral over-expression of the *Imp1-βgeo* fusion construct did not significantly affect the size or self-renewal of neurospheres (***Figure 2—figure supplement 2A–C***).

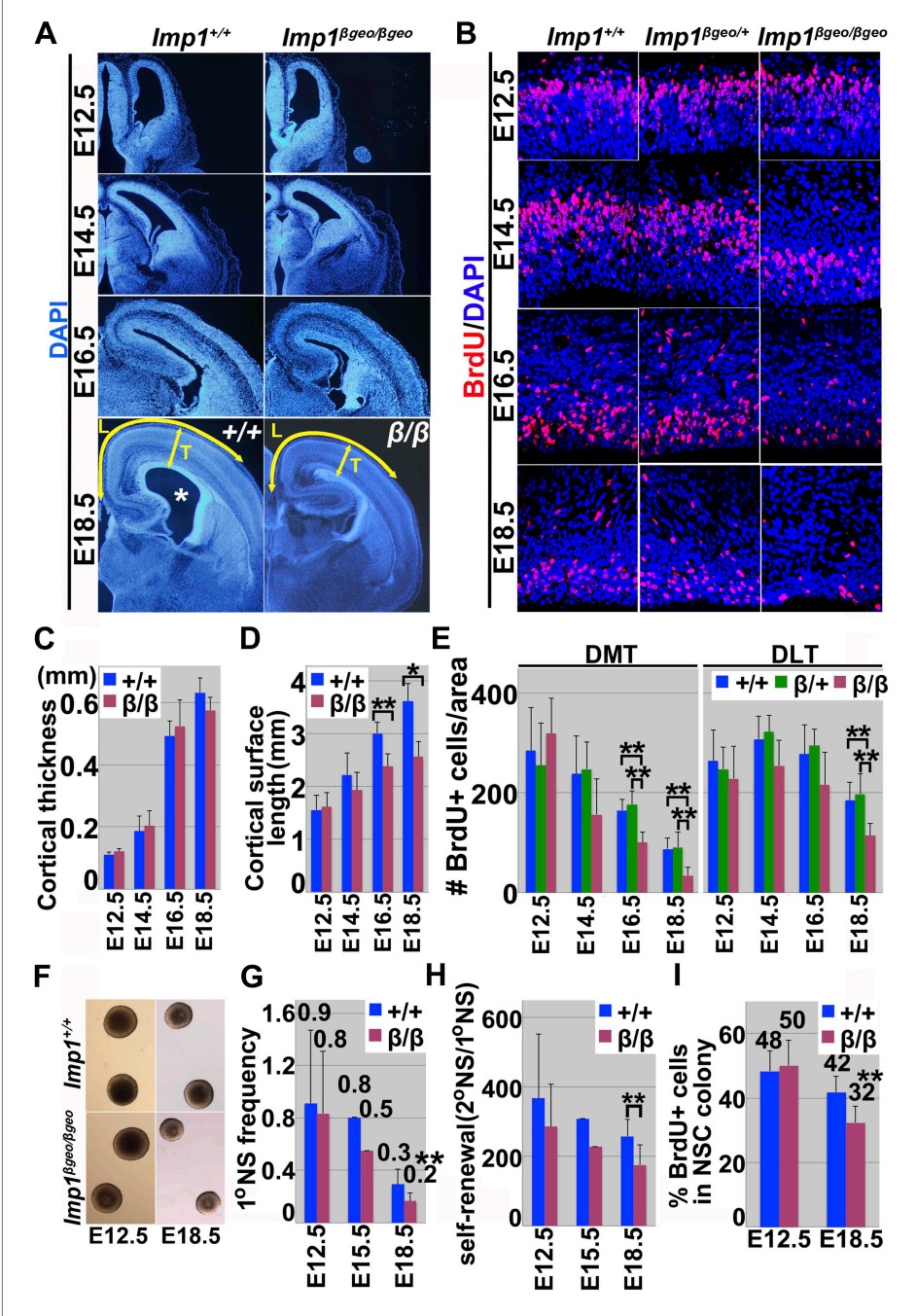

**Figure 2**. *Imp1* deficiency reduces brain size and pallial expansion due to reduced proliferation of fetal neural stem/progenitor cells. (**A**) Coronal sections of *Imp1⁺/⁺* and *Imp1^β-geo/ β-geo* telencephalons. The lateral ventricle is indicated with an asterisk in the *Imp1⁺/⁺* brain. Morphological abnormalities were visible in the *Imp1^β-geo/β-geo* telencephalon as early as at E14.5. Pallial expansion was impaired and the lateral ventricle collapsed in the *Imp1^β-geo/ β-geo* brain at E18.5. The length (L) and thickness (T) of the pallial regions are indicated with yellow arrows. (**B**) Dorsomedial telencephalon (DMT) sections from E12.5, E14.5, E16.5 or E18.5 *Imp1⁺/⁺ Imp1^β-geo/⁺* or *Imp1^β-geo/β-geo* embryos were stained with an antibody against BrdU. Reduction of cell proliferation was apparent in *Imp1^β-geo/ β-geo* telencephalon at E16.5 and E18.5. (**C–D**) The length (L) of the pial surface from the pallial/subpallial boundary to the retrosplenial cortex (L in panel **A**) was significantly shortened in the E16.5 and E18.5 *Imp1^β-geo/β-geo* telencephalon (\*\*p<0.05,\*p<0.01; four brains/genotype), but cortical thickness (T) was not significantly affected. (**E**) BrdU immunostaining revealed a significant reduction in the frequency of proliferating cells in E16.5 and E18.5 *Imp1^β-geo/β-geo*

*Figure 2. Continued on next page*

*Figure 2. Continued*

telencephalon but not at E14.5 or E12.5 (**p<0.05; mean ± SD for 3–5 brains/genotype/stage and 6–8 sections/brain). The reduction was more prominent in DMT, where *Imp1* was more strongly expressed, relative to dorsolateral telencephalon (DLT). (**F**) Typical neurospheres after 8 days culture from E12.5 or E18.5 dorsal telencephalon cells dissociated from wild-type or *Imp1$^{β-geo/β-geo}$* mice. (**G–H**) *Imp1* deficiency significantly reduced the percentage of cells that formed multilineage neurospheres and their self-renewal potential (the number of cells from individual primary neurospheres that formed multilineage secondary neurospheres upon subcloning), at E18.5 but not at E12.5 (**p<0.05; mean ± SD for 5–6 experiments/stage). We observed lower frequency and self-renewal potential of *Imp1* deficient multipotent neurospheres in two experiments performed at E15.5. (**I**) At E18.5, but not at E12.5, the percentage of cells within *Imp1$^{β-geo/β-geo}$* multilineage colonies that incorporated a 20 min pulse of BrdU was significantly lower than in *Imp1$^{+/+}$* colonies (three independent experiments/stage; **p<0.05).

The following figure supplements are available for figure 2:

**Figure supplement 1**. *Imp1*-deficient mice exhibit growth retardation and reduced brain mass but not increased cell death.

**Figure supplement 2**. *Imp1* deficiency, but not *Imp1-βgeo* overexpression, reduced fetal (but not adult) neural stem cell self-renewal.

Apoptotic cells were rare and their numbers were not affected by *Imp1* deficiency in the E13.5 or E17.5 telencephalon (*Figure 2—figure supplement 1E,F*). The reduced proliferation in the dorsal telencephalon appears to reduce pallial expansion in *Imp1$^{β-geo/β-geo}$* mice.

To assess whether this prenatal reduction in VZ cell proliferation in the *Imp1$^{β-geo/β-geo}$* telencephalon affected the self-renewal potential of individual neural stem cells we cultured cells from E12.5, E15.5, and E18.5 dorsomedial telencephalon from *Imp1$^{β-geo/β-geo}$* mice and littermate controls. We cultured the cells at low density in nonadherent cultures and then transferred individual neurospheres to adherent secondary cultures to determine the percentage of telencephalon cells that formed neurospheres that underwent multilineage differentiation. The percentage of cells that formed multipotent neurospheres did not differ between *Imp1$^{β-geo/β-geo}$* and wild-type telencephalon at E12.5, but was reduced in *Imp1$^{β-geo/β-geo}$* telencephalon at E15.5 and at E18.5 (*Figure 2G*). *Imp1$^{β-geo/β-geo}$* neurospheres did not significantly differ from wild-type neurospheres at E12.5, but were smaller than wild-type neurospheres at E15.5 and E18.5 and formed significantly (p<0.05) fewer multipotent secondary neurospheres upon subcloning (*Figure 2F,H*). This reduced self-renewal potential was associated with reduced proliferation within *Imp1$^{β-geo/β-geo}$* stem cell colonies at E15.5 and E18.5 (*Figure 2I*). As in vivo, cell death was rare within colonies of both genotypes (*Figure 2—figure supplement 1G*). The observation that IMP1 promotes the self-renewal of individual neural stem cells in culture demonstrates that IMP1 acts autonomously within neural stem/progenitor cells from the dorsal telencephalon to promote self-renewal.

We also cultured cells from the lateral ventricle SVZ of adult (P60) *Imp1$^{β-geo/β-geo}$* mice and littermate controls to assess whether *Imp1* deficiency affected the self-renewal of adult neural stem cells. Consistent with our inability to detect *Imp1* expression in these cells (*Figure 1—figure supplement 1D,F,G*), the percentage of cells that formed multipotent neurospheres, neurosphere size, and self-renewal potential did not significantly differ between *Imp1$^{β-geo/β-geo}$* and wild-type cells (*Figure 2—figure supplement 2D–F*).

## *Imp1* deficiency leads to premature differentiation in the telencephalon

To assess whether neural stem cell depletion was evident in the *Imp1$^{β-geo/β-geo}$* telencephalon in vivo, we examined the number of Pax6+ neural stem cells in the VZ of *Imp1$^{β-geo/β-geo}$* mice and littermate controls. The number of Pax6+ cells did not differ among genotypes at E12.5, but was significantly (p<0.05) reduced in the *Imp1$^{β-geo/β-geo}$* dorsomedial telencephalon from E14.5 to E18.5 and in the *Imp1$^{β-geo/β-geo}$* dorsolateral telencephalon at E18.5 (*Figure 3A,B*). The percentage of Pax6+ cells that were also BrdU+ or phospho-Histone H3+ (pH3+) did not differ between *Imp1$^{β-geo/β-geo}$* and control telencephalon at E12.5 or E14.5, but was significantly (p<0.05) reduced in the *Imp1$^{β-geo/β-geo}$* dorsomedial telencephalon at E18.5 (*Figure 3A,C*, *Figure 3—figure supplement 1A,B*). Similar to the overall reduction of cell proliferation (*Figure 2B,E*), the frequencies of Pax6+ cells and Pax6+BrdU+ cells

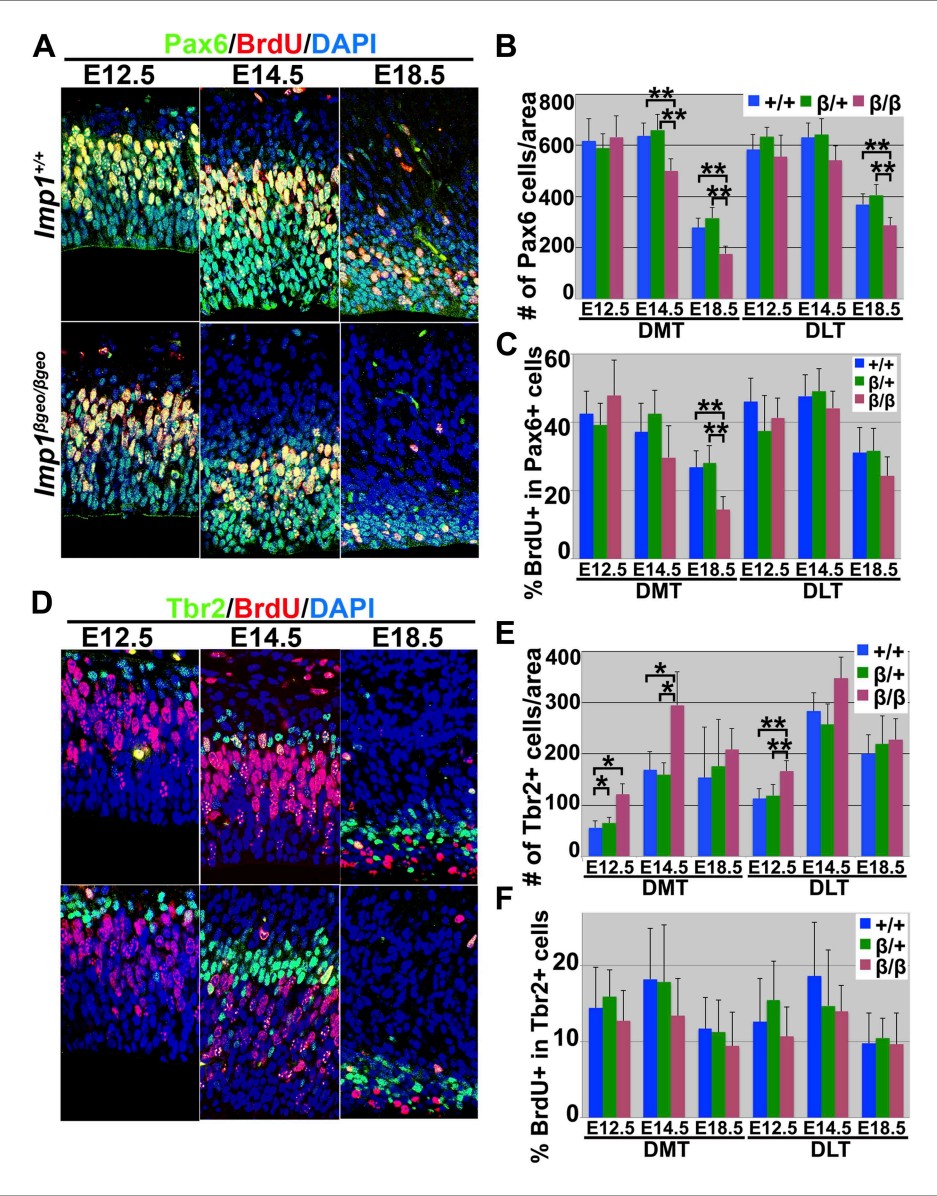

**Figure 3**. *Imp1* deficiency leads to precocious maturation of Pax6+ stem cells into Tbr2+ intermediate neuronal progenitors in the dorsal telencephalon. (**A**–**C**) *Imp1* deficiency significantly reduced the number of Pax6+ neural stem cells in the dorsomedial telencephalon (DMT) at E14.5 and E18.5, and in the dorsolateral telencephalon (DLT) at E18.5 (**p<0.05; mean ± SD for 3–4 mice/genotype at each stage with 6–8 sections/brain). *Imp1* deficiency significantly reduced the percentage of Pax6+ neural stem cells that were BrdU+ in E18.5 DMT (**p<0.05; mean ± SD for 3–4 mice/genotype at each stage with 6–7 sections/brain). (**D**–**F**) *Imp1* deficiency transiently increased the number of Tbr2+ intermediate progenitors in the DMT at E12.5 and E14.5, and in the DLT at E12.5 (*p<0.01; mean ± SD for 3–5 brains/genotype at each stage with 6–8 sections/brain). *Imp1* deficiency did not significantly affect the percentage of Tbr2+ cells that were also BrdU+ (mean ± SD for 3–4 mice/genotype with six sections/brain).

The following figure supplements are available for figure 3:

**Figure supplement 1**. *Imp1* deficiency significantly reduced the percentage of proliferating Pax6+ neural stem cells in E18.5 dorsomedial telencephalon (DMT).

were more strongly reduced in the dorsomedial telencephalon where *Imp1* is strongly expressed, as compared to the dorsolateral telencephalon (**Figure 3B,C**). We did not observe these effects in *Imp1*[β-geo/+] heterozygous mice (**Figure 3B,C,E,F**), demonstrating that they do not reflect a gain-of-function associated

with the mutant allele. Pax6+ neural stem cells therefore become depleted in the *Imp1*<sup>β-geo/β-geo</sup> dorsal telencephalon in vivo as a consequence of a loss of IMP1 function.

Complementary to the depletion of Pax6+ stem cells, the number of Tbr2+ intermediate neuronal progenitors was significantly ($p<0.01$) increased in E12.5 and E14.5 *Imp1*<sup>β-geo/β-geo</sup> dorsomedial telencephalon and in E12.5 *Imp1*<sup>β-geo/β-geo</sup> dorsolateral telencephalon as compared to littermate controls (*Figure 3D,E*). This increase in the number of Tbr2+ intermediate neuronal progenitors was transient as it was no longer statistically significant at E18.5 (*Figure 3D,E*). The increase in the number of Tbr2+ cells was not attributable to increased proliferation by these cells as the percentage of Tbr2+ cells that were also BrdU+ or pH3+ was not significantly different between *Imp1*<sup>β-geo/β-geo</sup> and control dorsal telencephalon at E12.5, E14.5, or E18.5 (*Figure 3D,F*, *Figure3—figure supplement 1C,D*). This suggests that Pax6+ stem cells are depleted in the absence of IMP1 by precocious maturation into Tbr2+ intermediate progenitors, transiently expanding the number of Tbr2+ cells from E12.5 to E14.5.

The number of Tuj1+ neurons was significantly ($p<0.05$) increased in the dorsomedial telencephalon of *Imp1*<sup>β-geo/β-geo</sup> mice at E12.5 and E14.5 relative to littermate controls (*Figure 4A,B*). The number of TAG-1+ corticofugal projection neurons was also significantly ($p<0.05$) increased in the dorsal telencephalon of *Imp1*<sup>β-geo/β-geo</sup> mice at E12.5 and E14.5 (not shown). We also dissociated cells from the telencephalons of *Imp1*<sup>β-geo/β-geo</sup> mice and littermate controls at E12.5 and cultured them adherently at clonal density. When colonies were stained after 9 days culture, we observed elevated numbers of Tuj1+ neurons within *Imp1*<sup>β-geo/β-geo</sup> multilineage colonies as compared to control multilineage colonies (*Figure 4C*). The frequency of neuron-only colonies formed by *Imp1*<sup>β-geo/β-geo</sup> telencephalon cells was also significantly ($p<0.05$) increased (*Figure 4D*) but the number of cells within these colonies was not affected (*Figure 4E*).

Although significant increases in the numbers of Tbr2+ progenitors and Tuj1+ neurons were observed beginning at E12.5 we did not observe a significant depletion of Pax6+ neural stem cells until E14.5. We believe this is because Pax6+ stem cells are much more numerous than Tbr2+ cells and Tuj1+ cells at E12.5. In wild-type mice we counted 617 Pax6+ cells as compared to only 55 Tbr2+ cells and 74 Tuj1+ cells in the same sections (*Figure 3B,E and 4B*). In *Imp1*-mutant mice we counted 633 Pax6+ cells as compared to 121 Tbr2+ cells and 117 Tuj1+ cells in the same sections. Therefore, the average increases in Tbr2+ and Tuj1+ cells totalled only 109 cells per section. Given that the standard deviation in Pax6+ cells per section was approximately 100, we were not able to detect such small changes in the number of Pax6+ cells at E12.5. At later stages of development when larger increases in Tbr2+ cells and Tuj1+ cells were observed we did detect significant declines in the numbers of Pax6+ cells.

To test whether *Imp1* deficiency also leads to premature gliogenesis, we examined telencephalon sections from *Imp1*<sup>β-geo/β-geo</sup> mice and littermate controls at E18.5. The number of GFAP+ astrocytes was significantly increased in the *Imp1*<sup>β-geo/β-geo</sup> telencephalon (*Figure 4F,G*), suggesting that gliogenesis is also precocious in *Imp1*<sup>β-geo/β-geo</sup> mice. We also adherently cultured cells from the E18.5 telencephalon of *Imp1*<sup>β-geo/β-geo</sup> mice and littermate controls at clonal density. When colonies were stained after 9 days of culture we observed increased numbers of GFAP+ cells within *Imp1*<sup>β-geo/β-geo</sup> neural stem cell colonies (*Figure 4H*). The frequency of glia-only colonies formed by *Imp1*<sup>β-geo/β-geo</sup> telencephalon cells was also significantly ($p<0.05$) increased (*Figure 4I*) but the number of cells within these colonies was not affected (*Figure 4J*). The observation that the effects of *Imp1* deficiency in vivo are greatest in the dorsal telencephalon where *Imp1* expression is highest, and the observation that these effects are also observed when cells are cultured at clonal density, suggest that IMP1 acts autonomously within neural stem cells in the dorsal telencephalon to prevent premature differentiation.

## Imp1 cell-autonomously promotes neural stem cell maintenance in dorsal telencephalon

To directly test whether IMP1 acts autonomously within neural stem cells, we injected virus bearing *Imp1* shRNA or scrambled control RNA into the telencephalic ventricles of wild-type mice in utero. Infection with the *Imp1* shRNA virus, but not the control virus, efficiently reduced IMP1 expression in neurospheres cultured from E14.5 telencephalon (*Figure 5A*). When these viruses were injected in the telencephalic ventricles of E14.5 wild-type mice in utero and analysed three days later, we distinguished infected from non-infected cells based on GFP expression (which was also carried in the viral vectors) in the dorsal telencephalon. GFP+ cells infected by the control shRNA tended to localize apically, among dividing Pax6+ cells in the VZ/SVZ (*Figure 5B*, *Figure 5—figure supplement 1*). In contrast, most GFP+ cells infected by the *Imp1* shRNA tended to localize basally, among Tbr2+ and Tuj1+ cells

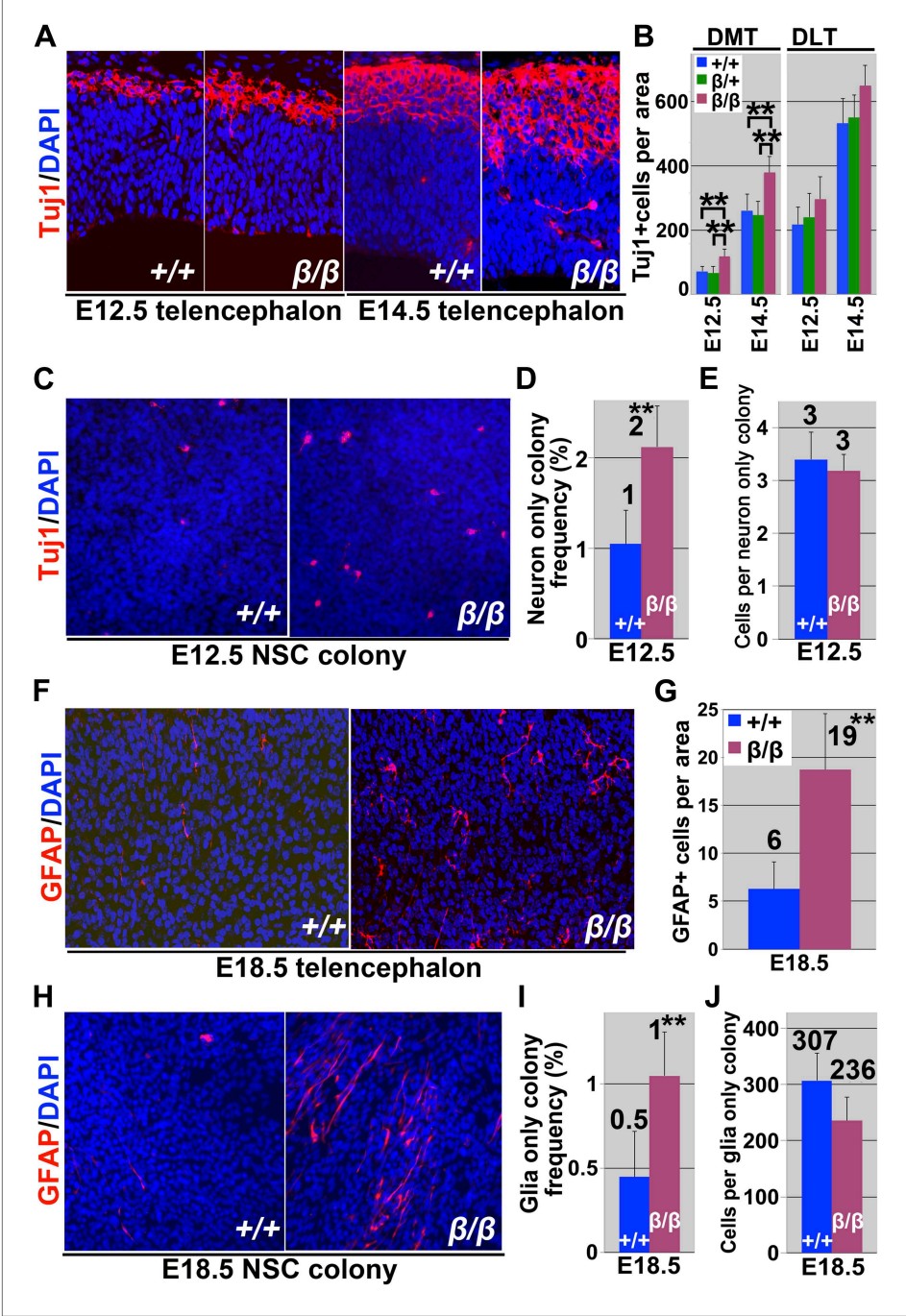

**Figure 4**. IMP1 prevents premature neuronal and glial differentiation by stem cells in the dorsal telencephalon. (**A**) Dorsomedial telencephalon sections from E12.5 or E14.5 control (+/+) or *Imp1*$^{β-geo/β-geo}$ (β/β) embryos were stained with an antibody against the neuronal marker Tuj1. (**B**) The number of Tuj1+ neurons per section was significantly increased in *Imp1*$^{β-geo/β-geo}$ (β/β) DMT as compared to littermate controls (+/+: *Imp1*$^{+/+}$ or β/+: *Imp1*$^{β-geo/+}$) at E12.5 and E14.5 (**p<0.05; mean ± SD for four brains/genotype with 4–6 sections/brain). (**C–E**) E12.5 dorsal telencephalon cells were cultured adherently for 9 days at clonal density. (**C**) Tuj1+ neurons were significantly more common in multipotent colonies from *Imp1*$^{β-geo/β-geo}$ (β/β) as compared to *Imp1*$^{+/+}$ (+/+) mice. (**D**) Significantly more neuron-only colonies were formed by *Imp1*$^{β-geo/β-geo}$ (β/β) as compared to *Imp1*$^{+/+}$ (+/+) telencephalon cells; however, the number of cells within control (+/+) or *Imp1*$^{β-geo/β-geo}$ (β/β) neuron-only colonies did not significantly differ (**E**; **p<0.05; mean ± SD for three independent experiments). (**F**) Dorsal telencephalon sections from E18.5 control (+/+) or *Imp1*$^{β-geo/β-geo}$ (β/β) embryos were stained with an antibody against GFAP. (**G**) The number of GFAP+
*Figure 4. Continued on next page*

*Figure 4. Continued*

astrocytes per section was significantly increased in the $Imp1^{\beta\text{-geo}/\beta\text{-geo}}$ (β/β) dorsal telencephalon as compared to littermate controls (+/+) at E18.5 (\*\*p<0.05; mean ± SD for four brains/genotype with 6–8 sections/brain). (**H–J**) E18.5 dorsal telencephalon cells were cultured adherently for 9 days at clonal density. (**H**) GFAP+ astrocytes were more common in multipotent colonies from $Imp1^{\beta\text{-geo}/\beta\text{-geo}}$ (β/β) as compared to $Imp1^{+/+}$ (+/+) mice. (**I**) Significantly more glia-only colonies were formed by $Imp1^{\beta\text{-geo}/\beta\text{-geo}}$ (β/β) as compared to $Imp1^{+/+}$ (+/+) telencephalon cells; however, the number of cells within control (+/+) or $Imp1^{\beta\text{-geo}/\beta\text{-geo}}$ (β/β) glia-only colonies did not significantly differ (**J**; \*\*p<0.05; mean ± SD for three independent experiments).

(*Figure 5B*, *Figure 5—figure supplement 1*). When cell proliferation was assessed by Ki67+ immunostaining, the percentage of GFP+ cells that were also Ki67+ was significantly (p<0.05) reduced in mice infected with *Imp1* shRNA as compared to control shRNA (*Figure 5C*, *Figure5—figure supplment 1*). This indicates that *Imp1* acts cell autonomously to maintain the proliferation of stem/progenitor cells in the dorsal telencephalon.

To assess whether *Imp1* knockdown affects the differentiation of neural stem cells, we injected viruses into the telencephalic ventricles of E12.5 wild-type mice in utero and 3 days later sections were immunostained with antibodies against Pax6, Tbr2 and Tuj1. In mice injected with *Imp1* shRNA, significantly fewer GFP+ cells were Pax6+ and significantly more GFP+ cells were Tbr2+ or Tuj1+ as compared to control shRNA (*Figure 5D–F*). These data indicate that *Imp1* acts cell autonomously to maintain Pax6+ stem cells in the dorsal telencephalon by opposing their maturation into Tbr2+ intermediate progenitors and their differentiation into neurons.

## Cell cycle exit of stem/progenitor cells is accelerated in the absence of *Imp1*

The timing of neuronal differentiation by stem/progenitor cells in the telencephalon is regulated by the timing of cell cycle exit, such that over-expression of cyclin D prolongs the proliferation of undifferentiated cells and delays the onset of neurogenesis (*Dehay and Kennedy, 2007*; *Lange et al., 2009*). We therefore examined the expression of cell cycle regulators in the dorsal telencephalon of $Imp1^{\beta\text{-geo}/\beta\text{-geo}}$ mice. We observed a significant (\*p<0.01; \*\*p<0.05) decline in the levels of all *cyclin D* family transcripts by both antibody staining (*Figure 6A*) and qPCR (*Figure 6B*) in the dorsomedial telencephalon of $Imp1^{\beta\text{-geo}/\beta\text{-geo}}$ mice as compared to control mice (*Figure 6A*). The decline in cyclin D1 staining was most pronounced in the dorsomedial telencephalon (*Figure 6A*, arrow) where *Imp1* expression was strongest (*Figure 6A*, arrowhead).

To assess whether increased neurogenesis in the *Imp1*-deficient telencephalon was associated with accelerated cell cycle exit, E14.5 $Imp1^{\beta\text{-geo}/\beta\text{-geo}}$ and littermate control mice were pulse labeled with BrdU for 24 hr then sacrificed and sections were stained with antibodies against BrdU and Ki67. The fraction of cells that exited the cell cycle during the BrdU pulse was estimated based on the frequency of BrdU+Ki67- cells and the fraction that remained in cycle was estimated based on the frequency of BrdU+Ki67+ cells, as described previously (*Chenn and Walsh, 2002*). The frequency of BrdU+Ki67- cells was significantly (p<0.05) higher in the $Imp1^{\beta\text{-geo}/\beta\text{-geo}}$ as compared to control telencephalon (*Figure 6C*), suggesting that cell cycle exit is accelerated in neural stem/progenitor cells from the $Imp1^{\beta\text{-geo}/\beta\text{-geo}}$ telencephalon. We could not detect a significant difference in the number of BrdU+Ki67+ cells. Most BrdU+Ki67+ cells were Pax6+ and most BrdU+Ki67- cells were either Tbr2+ or Tuj1+ (*Figure 6D*, *Figure 6—figure supplement 1*). These data suggest that most BrdU+Ki67+ cells were stem/progenitor cells while BrdU+Ki67- cells were a mixture of newborn neurons and neuronal progenitors.

Over-expression of *cyclin D1* or *cyclin D2* in neural stem/progenitor cells from E18.5 $Imp1^{\beta\text{-geo}/\beta\text{-geo}}$ telencephalon significantly increased Cyclin D1 or Cyclin D2 protein levels and proliferation, and reduced premature glial differentiation by $Imp1^{\beta\text{-geo}/\beta\text{-geo}}$ cells (*Figure 6E–G*). This suggests that reduced Cyclin D1 or Cyclin D2 expression contributes to the defects in *Imp1* deficient neural stem/progenitor cells. IMP1 is therefore required to maintain the expression of proteins that promote progression through G1 phase of the cell cycle such that *Imp1* deficiency accelerates cell cycle exit and differentiation.

## Canonical Wnt signaling promotes *Imp1* expression by neural stem/progenitor cells

During corticogenesis, a medial-lateral gradient of canonical Wnt signaling maintains the proliferation of undifferentiated Pax6+ stem cells in the pallial region of the telencephalon, preventing the premature

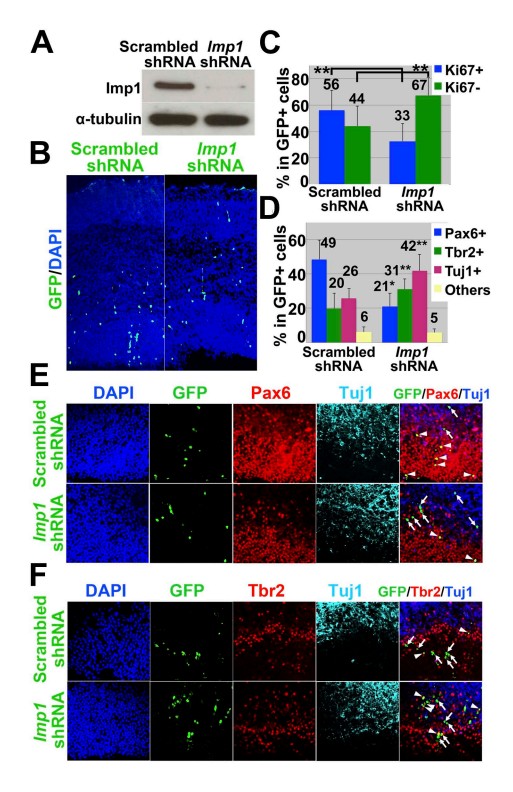

**Figure 5**. In utero knockdown of *Imp1* cell-autonomously reduces cell proliferation and accelerates differentiation of neural stem/progenitor cells. (**A**) Western blot of neurospheres cultured from E14.5 wild-type telencephalon cells infected with lentivirus bearing either control (scrambled) shRNA or *Imp1* shRNA. *Imp1* shRNA reduced IMP1 expression. (**B** and **C**) Viruses expressing either control or *Imp1* shRNA were injected into the telencephalic ventricles of E14.5 wild-type mice, infecting a small percentage of cells that could be identified based on GFP expression. Brains were fixed at E17.5 and dorsomedial telencephalon sections were immunostained with antibodies against GFP and Ki67. (**B**) Low magnification view of sections through the dorsal telencephalon including VZ (apical; bottom) and differentiated cell layers (basal; top). In *Imp1* shRNA infected telencephalon, GFP+ cells were more likely to be found basally as compared to control shRNA infected cells, suggesting that *Imp1* shRNA promoted the differentiation of infected cells. (**C**) The percentages of GFP+ (infected) cells that were Ki67+ (dividing) or Ki67- at E17.5. *Imp1* shRNA infection significantly reduced the percentage of GFP+ cells that were Ki67+ (**p<0.05; mean ± SD for four experiments). (**D–F**) Viruses expressing either control or *Imp1* shRNA were injected into the telencephalic ventricles of E12.5 wild-type mice. Brains were fixed at E15.5 and dorsomedial telencephalon sections were immunostained to assess the differentiation of GFP+ cells. (**D**) *Imp1* shRNA significantly increased the percentage of GFP+ cells that were Tbr2+ or Tuj1+ and significantly reduced

generation of Tbr2+ intermediate progenitors and differentiated neurons (***Chenn and Walsh, 2002***; ***Machon et al., 2007***; ***Wrobel et al., 2007***; ***Mutch et al., 2010***). The *Imp1* expression pattern we observed (***Figure 1D and 6A***, ***Figure 1—figure supplement 1B***) was reminiscent of the gradient of Wnt signaling observed in the telencephalon (***Machon et al., 2007***; ***Mutch et al., 2010***); furthermore, the *Imp1* loss-of-function phenotype in the telencephalon (***Figures 2–6***) was reminiscent of the phenotype observed in mutants with reduced Wnt signaling (***Mutch et al., 2010***). We therefore cultured E12.5 wild-type lateral telencephalon explants for 12 hr with or without recombinant Wnt3a, and examined *Imp1* transcript levels by qPCR. *Imp1* expression significantly (p<0.01) increased in the presence of Wnt3a (***Figure 6—figure supplement 2A***). Recombinant Dkk-1, which inhibits Wnt signaling, significantly (p<0.05) reduced *Imp1* expression (***Figure 6—figure supplement 2A***). Addition of BMP4, which also regulates dorsoventral patterning in the telencephalon (***Hebert et al., 2003***), did not affect *Imp1* expression (***Figure 6—figure supplement 2A***). These observations indicate that Wnt signaling can promote *Imp1* expression in neural stem/progenitor cells.

Next we assessed whether TCF4, a transcriptional mediator of canonical Wnt signaling, can directly bind to the *Imp1* enhancer/promotor. We did not detect TCF4 binding to exon 6 of *Imp1* or to the promoter/enhancer of *Lgi4* (negative controls) but did detect TCF4 binding to the first intron of *Axin* (a positive control [***Jho et al., 2002***]) (***Figure 6—figure supplement 2C***). We also examined two putative TCF/Lef binding sites that are conserved across species and located within 3 kb upstream of the *Imp1* translational start site (sites A and B in ***Figure 6—figure supplement 2B***). We detected significant enrichment of TCF4 binding at both sites by chromatin immunoprecipitation (***Figure 6—figure supplement 2C***). Relative luciferase activity was significantly reduced when one of these TCF/Lef binding sites (site B in ***Figure 6—figure supplement 2B***) was eliminated (***Figure 6—figure supplement 2D***). These observations suggest that canonical Wnt signaling might directly regulate *Imp1* expression.

To assess whether *Imp1* expression is regulated by canonical Wnt signaling under physiological conditions we examined the telencephalons of *Apc* mutant mice (*hGFAP-Cre; Apc^{flox/flox}*) to assess the consequences of increased Wnt signaling and *ß-catenin* mutant mice (*Nestin-Cre; Ctnnb1^{flox/flox}*) to assess the consequences of decreased Wnt signaling. *Imp1* expression was increased in the

*Figure 5. Continued*

the percentage that were Pax6+ (**p<0.05; mean ± SD for five experiments). (**E**) Triple immunostaining with antibodies against GFP, Pax6, and Tuj1. GFP+/Pax6+ cells (yellow cells in merged image) are marked with arrowheads, and GFP+/Pax6- cells are indicated with arrows. (**F**) Triple immunostaining with antibodies against GFP, Tbr2, and Tuj1. GFP+/Tbr2+ cells (yellow cells in merged image) are marked with arrowheads and GFP+/Tbr2- cells are indicated with arrows.

The following figure supplements are available for figure 5:

**Figure supplement 1**. *Imp1* knockdown in neural stem cells reduces cellular proliferation.

dorsal telencephalon of *Apc* deficient mice and decreased in the dorsal telencephalon of *ß-catenin* deficient mice relative to littermate controls (***Figure 7A,B***). Wnt signaling therefore promotes *Imp1* expression in the telencephalon in vivo.

To test whether there is a positive feedback between *Imp1* and canonical Wnt signaling (***Gu et al., 2008***) in neural stem/progenitor cells, we assessed ß-catenin expression in uncultured E13.5 telencephalon and in cultured neurospheres by qPCR and by western blot. However, we did not detect a significant difference in ß-catenin expression between *Imp1^{ß-geo/ß-geo}* and control cells (***Figure 6—figure supplement 2E,F***). We also did not detect a statistically significant difference in the stability of *ß-catenin* transcripts between *Imp1^{ß-geo/ß-geo}* and control neurospheres (***Figure 6—figure supplement 2G***), suggesting that there is no positive feedback from IMP1 to *ß-catenin* in neural stem/progenitor cells. IMP1 therefore appears to act downstream of Wnt signaling. The similarity between the *Imp1* expression pattern (***Figure 1D***) and the Wnt signaling gradient in the dorsal telencephalon (***Machon et al., 2007***; ***Mutch et al., 2010***), the observation that Wnt signaling promotes *Imp1* expression in the dorsal telencephalon in vivo (***Figure 7A***), and the similarity in the phenotypic consequences of reduced *Imp1* function (***Figures 2–6***) and reduced Wnt signaling (***Mutch et al., 2010***) all suggest that IMP1 acts autonomously within stem cells in the dorsal telencephalon to potentiate the effects of Wnt signaling during pallial expansion.

## *let-7* negatively regulates *Imp1* expression

To test whether *let-7* regulates *Imp1* expression in vivo we administered doxycycline for 5 days to wild-type and tetracyclin-inducible *let-7* transgenic (*ilet-7*) mice (***Zhu et al., 2011***). IMP1 protein expression was reduced in telencephalon cells freshly isolated from E13.5 *ilet-7* mice relative to controls, indicating that elevated expression of *let-7* negatively regulates IMP1 expression (***Figure 7C***). In E13.5 *ilet-7* telencephalon, the number of cells that incoporated a 1 hr pulse of BrdU and the number of Pax6+ neural progenitors were significantly reduced relative to controls, whereas the number of Tbr2+ neuronal progenitors and Tuj1+ neurons were significantly increased (***Figure 7D,E***). *let-7* induction thus phenocopied the IMP1 loss-of-function, consistent with the conclusion that *let-7* negatively regulates IMP1 expression.

To assess whether the elevated expression of *let-7* affects neural stem cell function we cultured dorsal telencephalon cells from doxycyclin-treated E13.5 wild-type and *ilet-7* mice. The percentage of cells that formed multipotent neurospheres, neurosphere size, and self-renewal potential were all significantly (p<0.01) reduced in *ilet-7+* cells relative to control cells (***Figure 7—figure supplement 1A–D***). These data indicate that increased *let-7* expression negatively regulates *Imp1* expression and fetal neural stem cell self-renewal.

Since there are 10 *let-7* family members that are thought to act redundantly to repress target gene expression (***Roush and Slack, 2008***), we used inducible *Lin28a* transgenic (*Lin28a Tg*) mice (***Zhu et al., 2010***) to test whether reduced expression of endogenous *let-7*s would increase IMP1 expression. Lin28a negatively regulates the expression of all mature *let-7* microRNAs (***Heo et al., 2008***; ***Newman et al., 2008***; ***Rybak et al., 2008***; ***Viswanathan et al., 2008***). We cultured E18.5 dorsal telencephalon cells or P60 lateral ventricle SVZ cells from doxycycline-treated wild-type or inducible *Lin28a Tg* mice and observed a significant reduction in endogenous *let-7b* expression in *Lin28a Tg* as compared to control neurospheres (p<0.01; sevenfold-reduction in *Lin28a Tg* E18.5 neurospheres and 14-fold reduction in P60 *Lin28a Tg* neurospheres; ***Figure 7—figure supplement 1E***). IMP1 protein expression was elevated in *Lin28a Tg* neurospheres relative to control neurospheres at E18.5 but not at P60 (***Figure 7F***). Neurosphere size and the self-renewal of multipotent neurospheres were significantly (*p<0.01) increased in E18.5 *Lin28a Tg* compared to control neurospheres (***Figure 7G–I***). These data suggest that physiological *let-7* expression reduces *Imp1* expression and self-renewal potential in fetal but not adult neural stem cells. Since we did not detect *Imp1* transcription or *Imp1^{ß-geo}* reporter expression

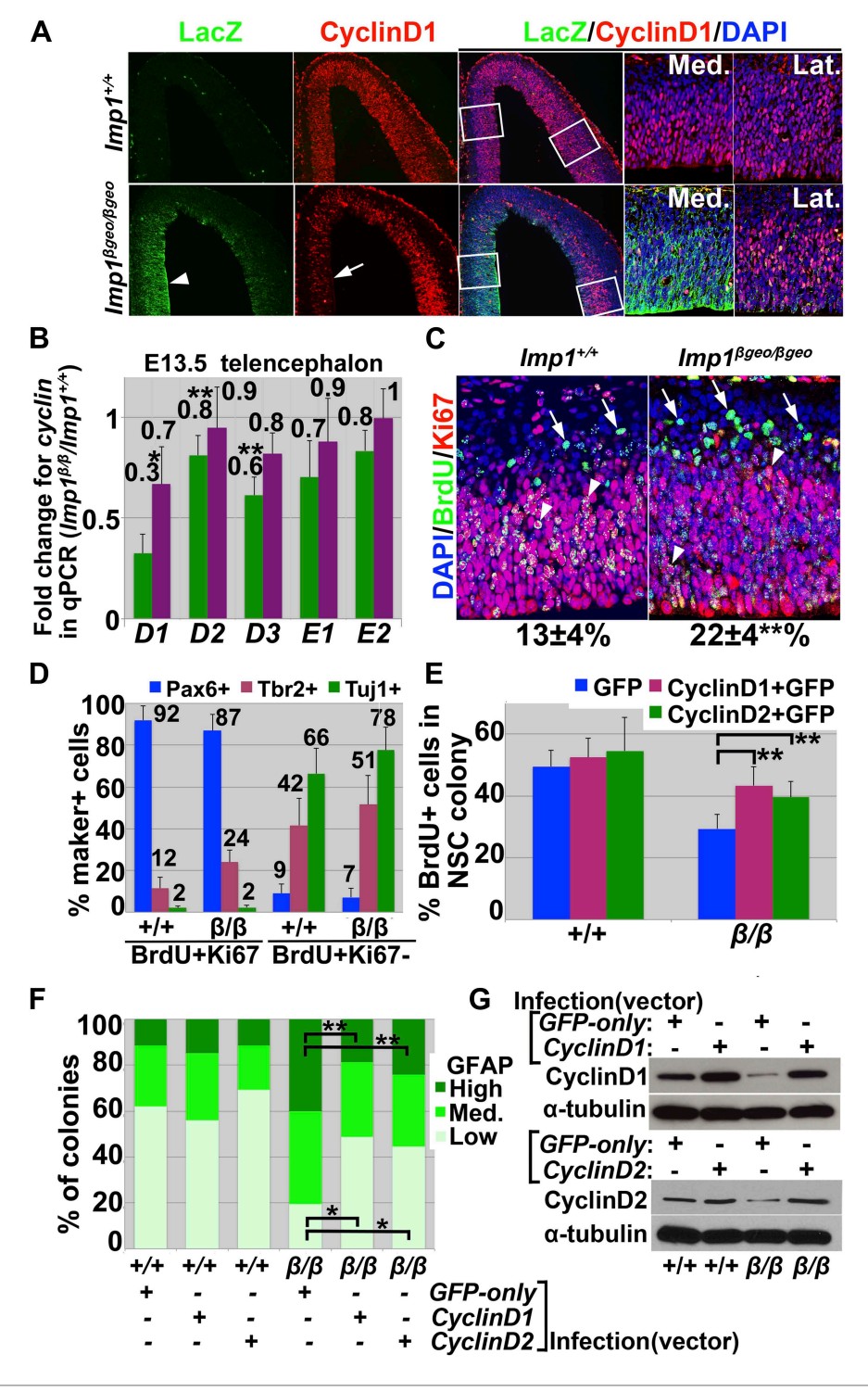

**Figure 6**. *Imp1* deficiency reduced cyclin D expression and accelerated cell cycle exit in the dorsal telencephalon. (**A**) Sections of E13.5 wild-type or *Imp1^{β-geo/β-geo}* dorsal telencephalon were immunostained with antibodies against Cyclin D1 and LacZ. Cyclin D1 expression was reduced relative to control in *Imp1^{β-geo/β-geo}* DMT (see arrow) where strong LacZ immunostaining indicated the highest levels of *Imp1* expression (see arrowhead). In contrast, Cyclin D1 immunostaining was retained in *Imp1^{β-geo/β-geo}* DLT, where lacZ immunostaining was weak. Higher magnification images on the right show boxed areas from low magnification images. (**B**) qPCR analysis of *cyclin D* and *cyclin E*

*Figure 6. Continued on next page*

*Figure 6. Continued*

transcripts in dorsomedial (green bar) and dorsolateral (purple bar) telencephalon from E13.5 mice. Each bar represents the fold change in $Imp1^{\beta\text{-}geo/\beta\text{-}geo}$/wild-type (error bars represent SD, four brains/genotype; *p<0.01, **p<0.05). (**C**) Dorsal telencephalon sections from E14.5 $Imp1^{+/+}$ or $Imp1^{\beta\text{-}geo/\beta\text{-}geo}$ mice that had been administered a single pulse of BrdU at E13.5 were stained with anti-BrdU and anti-Ki67 antibodies. Cells that exited the cell cycle after BrdU incorporation were BrdU+Ki67- (green; arrows) while cells that continued to divide were BrdU+Ki67+ cells (yellow; arrowheads). The $Imp1^{\beta\text{-}geo/\beta\text{-}geo}$ telencephalon had a significantly higher percentage of BrdU+Ki67- cells (22 ± 4%; **p<0.05; mean ± SD for four brains/genotype; 4–6 sections/brain). (**D**) Most BrdU+Ki67+ cells expressed Pax6 and most BrdU+Ki67- cells were either Tbr2+ or Tuj1+. Single channel images are presented in *Figure 6—figure supplement 1*. (**E–G**) E18.5 wild-type (+/+) or $Imp1^{\beta\text{-}geo/\beta\text{-}geo}$ (β/β) neural stem cells were infected with either *GFP-only* control retrovirus, *cyclin D1-GFP* retrovirus, or *cyclin D2-GFP* retrovirus and cultured. Within the resulting neural stem cell colonies, cell proliferation was assessed by BrdU incorporation (**E**), glial differentiation was assessed based on levels of GFAP staining (**F**), and Cyclin D1 or Cyclin D2 expression was examined by western blot (**G**). *Imp1* deficiency reduced Cyclin D1 or Cyclin D2 expression and neural stem cell proliferation and increased gliogenesis. These proliferation and premature gliogenesis phenotypes were partially rescued by cyclin D1 or cyclin D2 over-expression (three experiments; *p<0.01, **p<0.05).

The following figure supplements are available for figure 6:

**Figure supplement 1**. BrdU+/Ki67+ cells were Pax6+ while BrdU+/Ki67- cells were Tbr2+ or Tuj1+ in E14.5 dorsomedial telencephalon.

**Figure supplement 2**. Canonical Wnt signaling promotes *Imp1* expression.

---

in the postnatal forebrain (*Figure 1D,F*, *Figure 1—figure supplement 1D,F,G*), Imp1 expression is silenced postnatally by mechanisms independent of *let-7* (*Figure 7F*).

We also generated a floxed allele of the linked microRNAs *let-7b* and *let-7c2* (*let-7b/c2$^{fl}$*; *Figure 7—figure supplement 2*) and conditionally deleted it from fetal neural stem/progenitor cells using *Nestin-Cre* (*Nestin-Cre; let-7b/c2$^{flox/flox}$*). The percentage of cells that formed multipotent neurospheres, neurosphere size, and self-renewal potential did not significantly differ between *Nestin-Cre; let-7b/c2$^{flox/flox}$* mice and littermate controls at E18.5 or P60 (*Figure 7—figure supplement 1H–K*, and data not shown). Expression of *Imp1* and *Hmga2* transcripts also did not significantly differ between telencephalon cells obtained from *Nestin-Cre; let-7b/c2$^{flox/flox}$* mice and littermate controls (*Figure 7—figure supplement 1L*). This is consistent with the expectation that *let-7* family members act redundantly to regulate gene expression (*Roush and Slack, 2008*) such that deletion of *let-7b/c2* is not sufficient to change *Imp1* expression or neural stem cell function.

Although the major effects of Lin28a are mediated by *let-7*, there are also *let-7*-independent effects (*Cho et al., 2012*; *Wilbert et al., 2012*). To ensure that the effects of *Lin28a* on *Imp1* expression (*Figure 7F*) are really mediated by changes in the physiological levels of *let-7* family microRNAs we independently addressed this issue by testing whether *Imp1* expression is regulated by the *let-7* binding sites in the 3′ untranslated region (UTR). We overexpressed either full-lengh *Imp1* cDNA that contains all *let-7* binding sites (the *let-7* sensitive form) or a truncated *Imp1* cDNA that lacks the 3′-UTR (the *let-7* insensitive form) in E12.5 telencephalon cells or P60 SVZ cells. We were able to over-express either form of IMP1 in the E12.5 cells, when *let-7* expression is low, but in P60 SVZ cells higher levels of IMP1 were expressed from the truncated *Imp1* cDNA that lacks the *let-7* binding sites (*Figure 7J*). Ectopic expression of the *let-7* insensitive form of *Imp1*, but not the *let-7* sensitive form, significantly increased self-renewal and significantly reduced the number of differentiated neurons that arose in culture from neurospheres cultured from P60 SVZ (*Figure 7K*, *Figure 7—figure supplement 1F,G*). Over-expression of either form of *Imp1* did not significantly affect self-renewal in E12.5 telencephalon cells (*Figure 7K*). Regulation of IMP1 expression and neural stem cell function requires *let-7* binding sites in the *Imp1* 3′-UTR when *let-7* microRNA expression levels are elevated in vivo.

## *Imp1* promotes the expression of self-renewal genes, including *Hmga2*

To investigate the mechanism by which IMP1 promotes stem cell self-renewal we identified target RNAs bound by IMP1 in neural stem/progenitor cells. To do this we overexpressed FLAG-tagged IMP1 in neurospheres cultured from E13.5 $Imp1^{\beta\text{-}geo/\beta\text{-}geo}$ telencephalon then immunoprecipitated IMP1 and its target RNAs using an anti-FLAG antibody. When these RNAs were identified by deep-sequencing

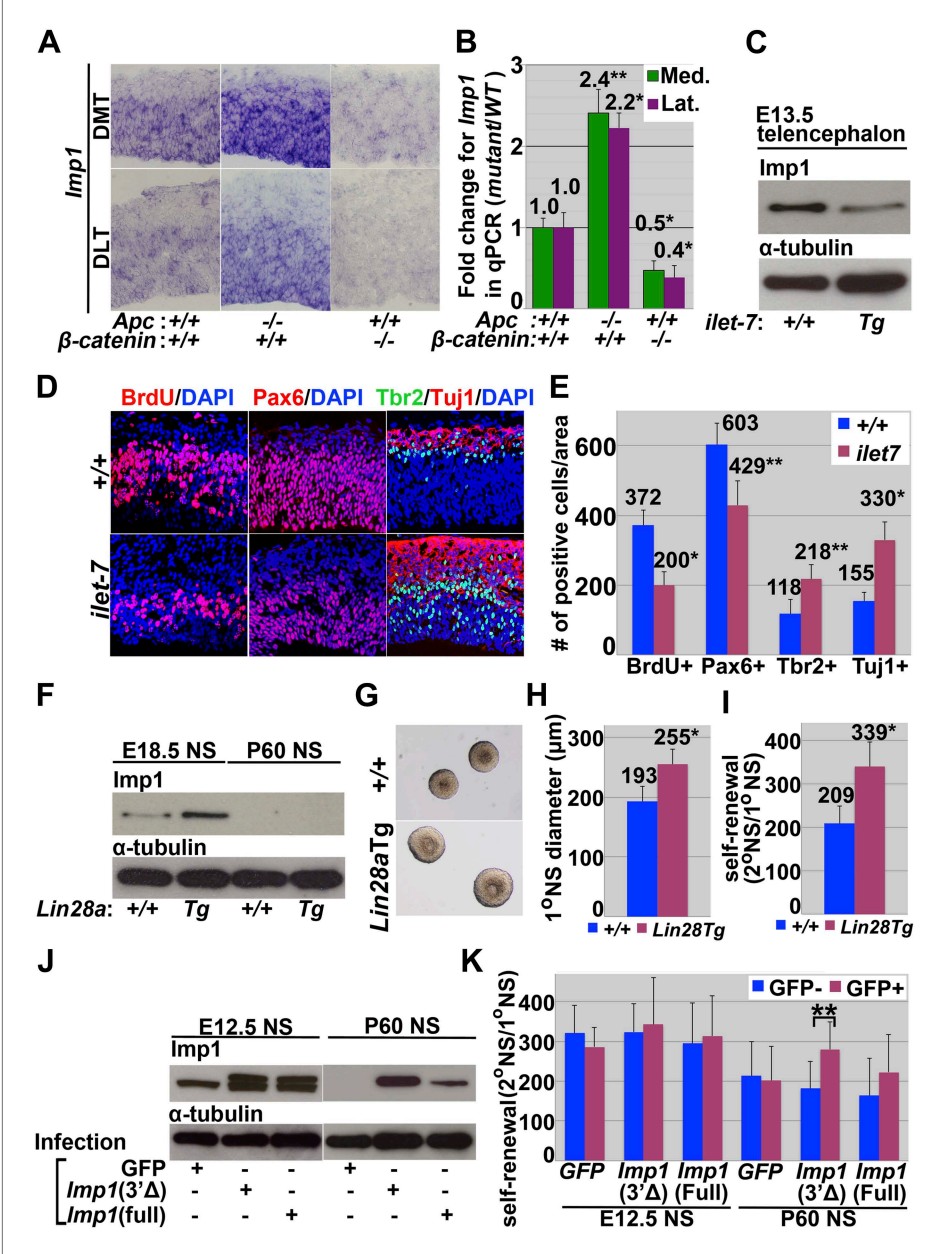

**Figure 7**. Canonical Wnt signaling promotes, and *let-7* inhibits, *Imp1* expression. (**A**–**B**) *Imp1* transcript levels were elevated in E14.5 dorsal telencephalon of *Apc*-deficient mice, and reduced in ß-catenin (*Ctnnb1*)-deficient mice by both in situ hybridization (**A**) and qPCR (**B**). Bars represent fold change in *Imp1* transcript levels in dorsomedial (green bars) and dorsolateral (purple bars) telencephalon of the indicated mutant mice/wild-type controls (*p<0.01, **p<0.05; error bar represents SD, 3–4 brains/genotype). (**C**) Western blot for IMP1 or α-tubulin in E13.5 telencephalon cells isolated from doxycycline administered wild-type (+/+) or *let-7* inducible transgenic mice (*ilet-7* Tg). (**D**–**E**) Dorsomedial telencephalon sections were prepared from doxycycline administered E13.5 wild-type (+/+) or *let-7* inducible transgenic mice (*ilet-7* Tg). Induction of *let-7* transgene expression significantly reduced the number of proliferating cells (assessed by a 1 hr pulse of BrdU) and the frequency of Pax6+ cells, and increased the numbers of Tbr2+ intermediate progenitors and Tuj1+ neurons (*p<0.01, **p<0.05; error bar represents SD, 3–4 mice/genotype). (**F**–**I**) E18.5 dorsal telencephalon cells or P60 SVZ cells were isolated from doxycycline administered wild-type (+/+) or *Lin28a* inducible transgenic mice (*Lin28a* Tg) and cultured as neurospheres. (**F**) Western blot for IMP1 or α-tubulin. IMP1 protein expression was elevated in *Lin28a* transgenic cells relative to control at E18.5 and not detected in either cells at P60. (**G**–**I**) *Lin28a* induction significantly increased the size of E18.5 neurospheres (**G** and **H**) and their self-renewal potential (**I**; *p<0.01; mean ± SD for four experiments). (**J**–**K**) E12.5 wild-type dorsal telencephalon *Figure 7. Continued on next page*

*Figure 7. Continued*

or P60 wild-type SVZ cells were infected with *GFP-only* control retrovirus (GFP), 3'UTR truncated 3XFLAG-*Imp1-GFP* retrovirus (3Δ), or full length 3XFLAG *Imp1-GFP* retrovirus (full). Truncated *Imp1* lacked *let-7* binding sites in the 3' UTR. (**J**) Over-expression of truncated 3XFLAG-*Imp1-GFP* increased IMP1 protein expression more efficiently than full length 3XFLAG *Imp1-GFP* over-expression at P60 when *let-7* expression is high. The smaller band corresponds to endogenous IMP1 while the larger band corresponds to FLAG-tagged IMP1. (**K**) Only 3' UTR truncated 3XFLAG *Imp1-GFP* over-expression significantly increased the self-renewal of neurospheres relative to uninfected cells at P60.

The following figure supplements are available for figure 7:

**Figure supplement 1**. *let-7* over-expression inhibits neural stem cell self-renewal while *Imp1* over-expression inhibits neurogenesis.

**Figure supplement 2**. Generation of a conditional mutant allele of *let-7b* and *let-7c2* (*let-7b/c2ᶠ*) by gene targeting.

we found transcripts from 103 genes that were significantly enriched in the immunoprecipitated fraction compared with total RNA (three samples per genotype, >twofold-enrichment, p<0.05) (**Supplementary file 1A**). Many of these gene products are associated with differentiated cells and their expression increased in the telencephalon between E13.5 and E18.5 (as IMP1 expression declined) (**Figure 8—figure supplement 1A**) and in cultured neurospheres as differentiation progressed (**Figure 8—figure supplement 1B**). We could not detect a significant change in the expression of these genes by qPCR in *Imp1*-deficient neurospheres (**Figure 8A**), but did observe increased expression of several proteins against which effective antibodies were available (**Figure 8B**). We also detected a significant shift of these transcripts to the polysomal fraction in *Imp1*-deficient neurospheres (**Figure 8C**, **Figure 8—figure supplement 1C**). These data suggest that IMP1 acts post-transcriptionally to negatively regulate the expression of some proteins associated with neural differentiation.

Two *let-7* microRNA targets that promote neural stem cell self-renewal, *Hmga2* and *cyclin D2* (*Ccnd2*), were also significantly enriched in the fraction bound by IMP1 (**Supplementary file 1A**). *Ccnd2* expression was reduced in the dorsomedial telencephalon of *Imp1* deficient mice (**Figure 6B**). To assess whether IMP1 regulates *Hmga2* expression we compared the levels of *Hmga2* and it's family member *Hmga1* by qPCR in E13.5 and E18.5 dorsomedial and dorsolateral telencephalon cells from *Imp1^{β-geo/β-geo}* mice and littermate controls. *Hmga2*, but not *Hmga1*, transcript levels were significantly (p<0.05) reduced in *Imp1^{β-geo/β-geo}* cells by qPCR (**Figure 8D**). This reduction in *Hmga2* expression was also confirmed at the protein level by western blot (**Figure 8E**). In sections from *Imp1^{β-geo/β-geo}* mice, the reduction in *Hmga2* expression was confirmed by in situ hybridization in the dorsomedial and dorsolateral telencephalon (where *Imp1* is normally expressed) but not in the ventral telencephalon (where *Imp1* is not normally expressed; compare **Figure 8F** to **Figure 1—figure supplement 1B,E**). Indeed, the region of the dorsal telencephalon in which *Hmga2* expression declined (**Figure 8F**) corresponded precisely with the region of *Imp1* expression (**Figure 1D**). This demonstrates that IMP1 acts autonomously within stem cells in the dorsal telencephalon to maintain HMGA2 expression.

IMP1 appears to promote HMGA2 expression by increasing the stability of its mRNA as we detected a significantly (p<0.05) accelerated decay of mRNA for *Hmga2*, but not *Hmga1*, in neurospheres cultured from E13.5 *Imp1^{β-geo/β-geo}* mice as compared to wild-type controls (**Figure 8G**). HMGA2 did not appear to regulate *Imp1* expression because *Imp1* transcript levels were not affected by *Hmga2* deficiency by either qPCR (**Figure 8H**) or in situ hybridization (**Figure 8I**).

We previously demonstrated that HMGA2 promotes neural stem cell self-renewal in the fetal telencephalon, partly by negatively regulating the expression of *Ink4a*, which encodes a cyclin-dependent kinase inhibitor (**Nishino et al., 2008**). Consistent with this, we detected significantly (p<0.05) increased *Ink4a* expression in neurospheres cultured from E18.5 *Imp1^{β-geo/β-geo}* mice as compared to wild-type controls (**Figure 8J**). Over-expression of *Hmga2* restored normal levels of Hmga2 protein in E18.5 *Imp1^{β-geo/β-geo}* neurospheres (**Figure 8K**) and significantly increased their self-renewal potential (**Figure 8L**; p<0.05). IMP1 therefore promotes the self-renewal of fetal neural stem cells partly by promoting the expression of HMGA2. This suggests that part of the mechanism by which neural stem cells expand in the dorsal telencephalon in response to Wnt signaling is through IMP1-promoted HMGA2 expression,

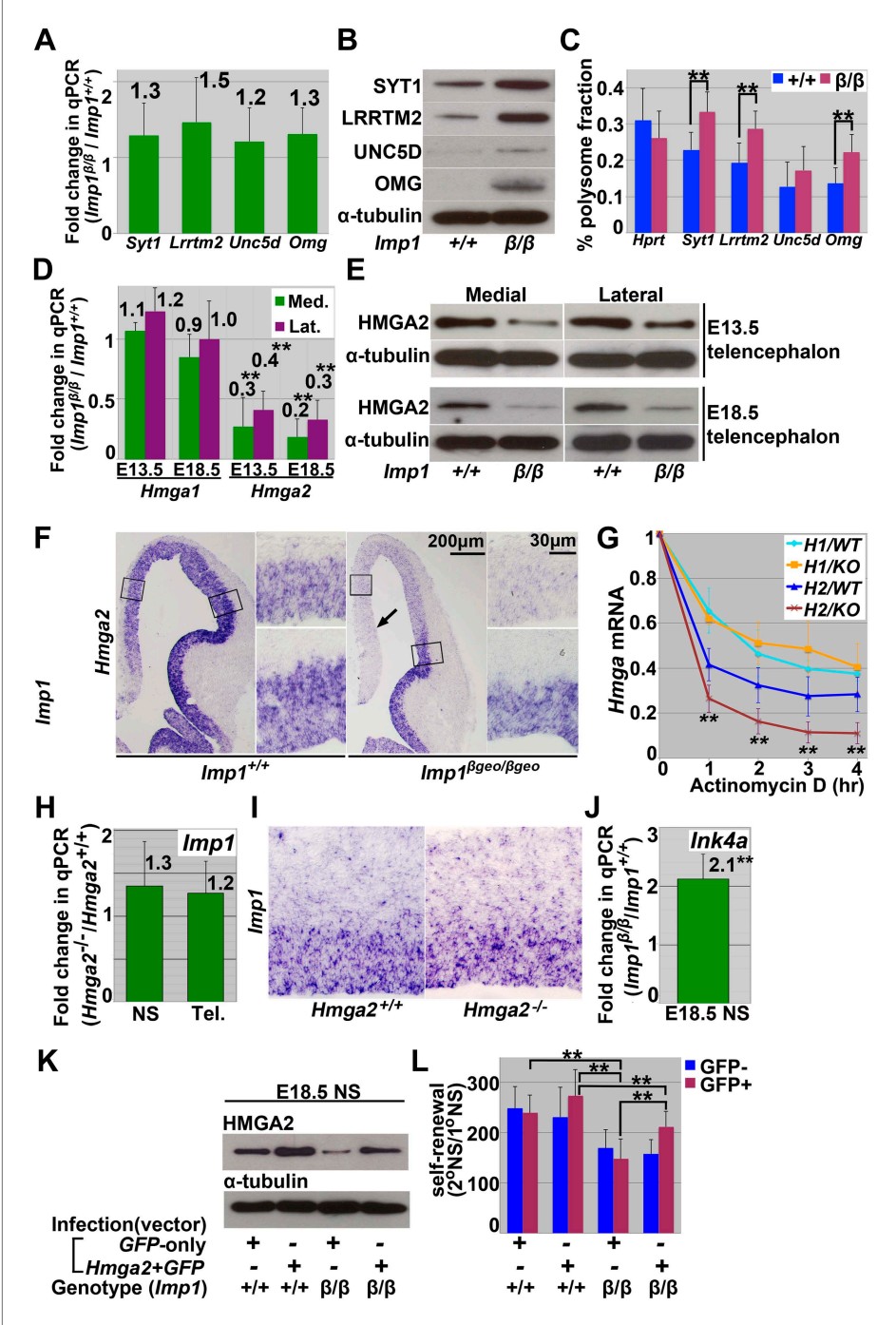

**Figure 8**. *Imp1* acts post-transcriptionally and cell autonomously to negatively regulate the expression of gene products associated with differentiation and to promote the expression of self-renewal genes, including *Hmga2*. (**A**–**B**) *Imp1* deficiency did not affect the levels of *synaptotagmin1* (*Syt1*), *Leucine rich repeat transmembrane neuronal 2* (*Lrrtm2*), *Unc5d*, or *Oligodendrocyte myelin glycoprotein* (*Omg*) transcripts by qPCR in E13.5 dorsal telencephalon-derived neurospheres; however, *Imp1* deficiency did increase the levels of SYT1, LRRTM2, UNC5d, and OMG proteins. (**C**) Neurosphere lysates from E13.5 wild-type (+/+) or *Imp1*β-geo/β-geo (β/β) dorsal telencephalon were fractionated in a 10–50% sucrose gradient (*Figure 8—figure supplement 8*) and the transcripts in the polysome fraction were assessed by qPCR. *Synaptotagmin1* (*Syt1*), *Leucine rich repeat transmembrane neuronal 2* (*Lrrtm2*), and *Oligodendrocyte myelin glycoprotein* (*Omg*) transcripts were significantly enriched in the polysome fractions in *Imp1* deficient cells relative to control cells (**p<0.05; error bars represent SD from three independent experiments). *Figure 8. Continued on next page*

*Figure 8. Continued*

(**D**) *Hmga2,* but not *Hmga1,* transcript levels were significantly reduced in *Imp1^{β-geo/β-geo}* dorsomedial (green bar) and dorsolateral telencephalon (purple bar) (*p<0.01, **p<0.05; error bars represent SD, 3–4 brains/genotype). (**E**) *Imp1* deficiency reduced HMGA2 levels in dorsomedial, and dorsolateral telencephalon. (**F**) *Imp1* deficiency reduced *Hmga2* transcript levels (by in situ hybridization; purple) in dorsomedial and dorsolateral telencephalon from E13.5 mice. Higher magnification views of the boxed regions of dorsomedial (upper) or dorsolateral (lower) telencephalons are shown to the right of lower magnification images. Note that *Hmga2* expression declined to a greater extent in the dorsomedial telencephalon (arrow) where *Imp1* expression is highest and did not decline in the ventral telencephalon where *Imp1* is not expressed at this stage (see **Figure 1D**). *Imp1* is thus required cell autonomously in the dorsal telencephalon to maintain *Hmga2* expression. (**G**) *Hmga2,* but not *Hmga1,* transcript levels were significantly (**p<0.05) reduced after Actinomycin D treatment in *Imp1^{β-geo/β-geo}* (red) relative to wild-type (blue) neurospheres cultured from E13.5 dorsal telencephalon. Error bars represent SD in three experiments. (**H**–**I**) *Hmga2* deficiency did not affect *Imp1* transcript levels in E13.5 telencephalon cells by qPCR (**H**) or in situ hybridization (**I**) (three independent experiments). (**J**) *Ink4a* transcript levels were significantly elevated by *Imp1* deficiency in neurospheres cultured from E18.5 wild-type or *Imp1^{β-geo/β-geo}* dorsal telencephalon (**p<0.05; error bars represent SD, 3–4 mice/genotype). (**K** and **L**) E18.5 wild-type (+/+) or *Imp1^{β-geo/β-geo}* (β/β) dorsal telencephalon cells were infected with *GFP-only* control retrovirus (GFP) or *Hmga2-GFP* retrovirus. *Imp1* deficiency reduced HMGA2 expression and neural stem cell self-renewal but both were restored by *Hmga2* over-expression.

The following figure supplements are available for figure 8:

**Figure supplement 1**. Multiple mRNAs bound by IMP1 increased their expression during brain development and neural differentiation.

---

but that HMGA2 expression and neural stem cell expansion decline postnatally as a consequence of increased *let-7* expression (**Nishino et al., 2008**) and a loss of IMP1 expression (**Figure 1A**).

## Discussion

IMP1 is one element of a network of heterochronic genes that regulates temporal changes in neural stem cell function throughout life (see **Figure 9** for a graphical summary). Expression of the *let-7* target, *Imp1,* by stem cells in the dorsal telencephalon (**Figure 1**, **Figure 1—figure supplements 1B,E and 2**) promoted the expansion of undifferentiated stem cells in response to Wnt signaling during fetal development. Wnt signaling promoted *Imp1* expression in the dorsal telencephalon in a medial-lateral gradient (**Figure 7A–B**, **Figure 6—figure supplement 2A–D**) similar to the gradient in canonical Wnt signaling (**Machon et al., 2007**; **Mutch et al., 2010**). *Imp1* deficiency reduced stem cell self-renewal potential (**Figure 2F–I**) and caused premature neuronal and glial differentiation (**Figure 4**), leading to stem cell depletion (**Figures 2G, 3A–C**) and reduced pallial expansion (**Figure 2A,D**). Therefore, IMP1 expression by stem/progenitor cells during forebrain development regulates the timing of neuronal and glial differentiation. The postnatal loss of IMP1 expression may contribute to the decline in neural stem cell function during adulthood.

As *let-7* expression increases during late fetal development (**Figure 1B**), *Imp1* expression declines (**Figure 1D**). Increasing *let-7* expression in neural stem cells reduces IMP1 expression (**Figures 1C and 7C**), the number of neural stem cells, and their self-renewal potential (**Figure 7D–E**, **Figure 7—figure supplement 1A–D**). Reducing the expression of mature *let-7* microRNAs by inducing *Lin28a* expression increases IMP1 expression and neural stem cell self-renewal (**Figure 7F–I**). The *let-7* binding sites in the *Imp1* 3′UTR impede IMP1 expression in adult neural stem cells (**Figure 7J–K**). Nonetheless, *let-7* is not solely responsible for the perinatal loss of *Imp1* expression as *β-geo* expression in *Imp1^{β-geo/+}* mice is also lost perinatally even though *β-geo* does not carry the *Imp1* 3′ UTR that contains the *let-7* binding sites. Increasing expression of *let-7* microRNAs during fetal development reduces IMP1 expression before other mechanisms silence *Imp1* postnatally.

IMP1 is not detectably expressed in the adult forebrain (**Figure 1A**, **Figure 1—figure supplement 1**) and reduced *let-7* microRNA expression does not restore IMP1 expression in adult neural stem cells (**Figure 7F**). IMP1 is also dispensable for the regulation of self-renewal in adult neural stem cells (**Figure 2—figure supplement 2D–F**). IMP1 is therefore required for the regulation of fetal but not adult neural stem/progenitor cells and contributes to the increased proliferation of undifferentiated cells in the fetal as compared to the adult forebrain. *let-7* regulates the function of stem cells in the fetal and adult forebrains (**Zhao et al., 2010**); however, *let-7* microRNAs appear to regulate different

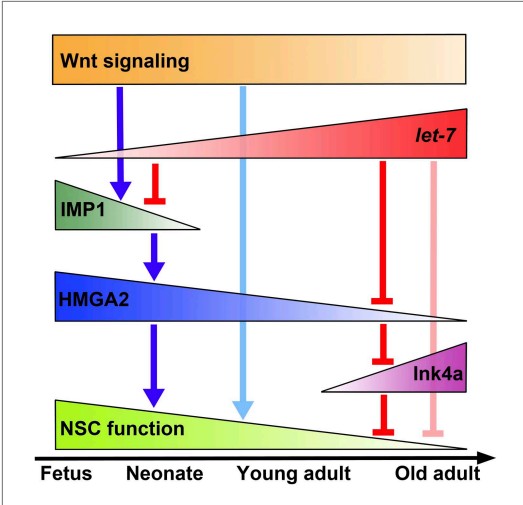

**Figure 9**. Schematic showing a network of heterochronic genes that regulate temporal changes in CNS stem cell properties from fetal development throughout adulthood. A network of heterochronic genes changes with age, leading to temporal changes in stem cell properties from fetal development throughout adulthood. The promotion of expression/function is indicated as blue arrows and negative regulation is indicated as red bars. In fetal neural stem cells, *Imp1* expression is promoted transcriptionally by canonical Wnt signaling and inhibited post-transcriptionally by *let-7*. IMP1 expression is reduced by increasing *let-7* expression in late fetal development and transcriptionally silenced postnatally. HMGA2 expression is high in early development but declines with time in response to declining IMP1 and increasing *let-7*. Increasing *let-7* expression contributes to the reduction in HMGA2 expression over time during adulthood. HMGA2 negatively regulates the expression of Ink4a/Arf. Declining expression of HMGA2 during adulthood allows the expression of these gene products to increase during aging. Overall, neural stem cell function declines over time. A decline in Wnt signaling during aging also contributes to these effects (*Seib et al., 2013*).

targets at different ages in neural stem cells. *let-7* microRNAs, IMP1, and HMGA2 are three important elements of a network of heterochronic genes that regulates temporal changes in stem cell function throughout life.

*Imp2* and *Imp3* exhibit similar expression patterns as *Imp1* (*Figure 1—figure supplement 1H,I*). The overlap in expression among IMP family members suggests potential redundancy among family members in stem cells from the dorsal telencephalon. This raises the question of whether compound deletion of multiple family members would further accelerate stem cell depletion.

IMP1 is known to increase the levels of some proteins by increasing the stability of their mRNAs (*Noubissi et al., 2006*) and to reduce the levels of other proteins by inhibiting translation (*Hansen et al., 2004*; *Atlas et al., 2007*). Consistent with this, IMP1 post-transcriptionally reduced the levels of some proteins associated with differentiation (*Figure 8A–C*, *Figure 8—figure supplement 1A–C*) while promoting the expression of the self-renewal factor, HMGA2 (*Figure 8D–F*). IMP1 appeared to promote HMGA2 expression by increasing the half-life of *Hmga2* mRNA (*Figure 8G*), leading to higher levels of HMGA2 (*Figure 8E*) and lower levels of $p16^{Ink4a}$ (*Figure 8J*). We showed previously that HMGA2 promotes, and $p16^{Ink4a}$ inhibits, stem cell self-renewal in the telencephalon (*Nishino et al., 2008*). The postnatal lack of IMP1 expression in forebrain neural stem cells (*Figure 1D, F*, *Figure 1—figure supplement 1D,F,G*) is therefore likely to contribute to their reduced HMGA2 expression and self-renewal potential.

Recently, two genome-wide studies in HEK293 cells suggested the existence of putative IMP1 recognition sequences: 5'-CAUH-3' (H = A, U, or C) (*Hafner et al., 2010*) and 5'-CCYHHCC-3' (Y = C, U and H = A, C, U) (*Jønson et al., 2007*). We found these sequences at least once in all of our IMP1 pulled-downed mRNAs, but since these sequences would be expected to occur by chance in the mRNAs, no conclusion can be drawn regarding the functional relevance of these binding sites in the mRNAs we observed.

Recently, *Imp* (a fly ortholog of mammalian *Imp1*) was shown to non-cell-autonomously regulate the aging of germline stem cells in the fly testis (*Toledano et al., 2012*). This raises the possibility that IMP1 might have non-cell-autonomous effects on mammalian stem cells, at least in certain contexts. Indeed, the overall growth retardation observed in *Imp1*-mutant mice (*Figure 2—figure supplement 1A*) could reflect non-cell-autonomous effects of *Imp1* deficiency. Nonetheless, our data indicate that IMP1 cell-autonomously regulates neural stem cell function in the dorsal telencephalon. First, neural stem cell specific knockdown of *Imp1* in a small percentage of cells in the developing telencephalon reduced cell proliferation and accelerated differentiation in a cell autonomous way (*Figure 5*) .Second, all of our experiments demonstrating neural stem/progenitor cell phenotypes in *Imp1* deficient (*Figures 2F–I and 4*), *Lin28* transgenic (*Figure 7G–I*) or *ilet-7* transgenic mice (*Figure 7—figure supplement 1A–D*) involved studies of individual isolated neural stem cells in culture. Third, in all of our experiments that involved viral infection (*Figure 2—figure supplement 2B,C*, *Figures 6E–F, 7K, 8L*,

*Figure 7—figure supplement 1F–G*), we compared the growth of infected neurospheres to non-infected neurospheres growing in the same cultures, or in control cultures. We never observed non-cell autonomous effects of infected neurospheres on non-infected neurospheres within the same cultures. Finally, we observed a correlation between *Imp1* expression and function within the telencephalon: the phenotypes observed in *Imp1* deficient mice were consistently more profound in the dorsomedial telencephalon, where *Imp1* expression was high, than in the dorsolateral telencephalon, where *Imp1* expression was low (*Figures 2E and 3*, *Figure 4B*). IMP1 therefore cell-autonomously regulates the function of neural stem/progenitor cells in the dorsal telencephalon, but likely has non-cell-autonomous effects in other contexts.

We have thus demonstrated a novel function of the *let-7* microRNA target, IMP1, to promote the expansion of neural stem cells during cortical development. *let-7b* also negatively regulates the self-renewal of adult neural stem cells by reducing the expression of *Hmga2* (*Nishino et al., 2008*) and *TLX* (*Zhao et al., 2010*). This is consistent with our conclusion that a network of *let-7* gene targets regulates stem cell function and that the regulation of different targets at different ages by *let-7* contributes to temporal changes in stem cell properties.

Other RNA binding proteins also regulate stem cell function. FBF proteins and GLD proteins control germline stem cell maintenance and the timing of meiosis in *C. elegans* (*Kimble and Crittenden, 2007*). The evolutionarily conserved Piwi family proteins that bind to Piwi-interacting RNAs are also required for the maintenance of germline stem cells (*Juliano et al., 2011*). Musashi proteins appear to promote the self-renewal of fetal and adult stem cells from multiple tissues by translational repression of target RNAs (*Sakakibara et al., 2002*; *Okano et al., 2005*; *Kharas et al., 2010*). Lin28 is preferentially expressed by embryonic cells and essential for the development of primordial germ cells (*West et al., 2009*; *Viswanathan and Daley, 2010*). Indeed, the decline in Lin28 expression with time during fetal development may contribute to the increase in *let-7* expression and the decline in IMP1 expression.

A network of heterochronic genes including *let-7* microRNAs, *Imp1*, *Hmga2*, and the cell cycle regulators *cyclin D* and *p16^{Ink4a}*, regulates temporal changes in stem cell function. While we have demonstrated the functional importance of this network in neural stem cells, this network is also likely to regulate temporal changes in stem cells from other tissues. Beyond the network components we identified, many additional *let-7* target genes are likely to regulate developmental changes in stem cells, integrating stem cell properties with changing tissue growth and regeneration demands throughout life.

## Materials and methods

### Mice

*Imp1^{β-geo/+}* (*Hansen et al., 2004*) (MMRRC stock number 011720-UCD), *APC^{flox/+}* (*Shibata et al., 1997*), *β-catenin^{flox/+}* (*Brault et al., 2001*), *human-GFAP Cre* (*Malatesta et al., 2003*), *Lin28a* transgenic (*Zhu et al., 2010*), *ilet-7* transgenic (*Zhu et al., 2011*), *Nestin-Cre* (*Tronche et al., 1999*), *Hmga2^{+/−}* mice (*Zhou et al., 1995*), and *let-7b/c2^{fl/+}* mice were each backcrossed at least six times onto a C57BL/Ka background and housed at the University of Michigan Unit for Laboratory Animal Medicine or the University of Texas Southwestern Medical Center Animal Resource Center. Mice were genotyped by PCR. For *let-7* or *Lin28* induction, mice were administered water containing 2 µg/ml doxycycline (Research Products International Co., Mount Prospect, IL) for 4 to 5 days.

### Cell culture and self-renewal assay

CNS progenitors were isolated as described in prior studies (*Molofsky et al., 2005*; *Nishino et al., 2008*). For adherent cultures, CNS progenitors were plated at a clonal density of 0.66 cells/µl (1000 cells per 35 mm well) in 6-well plates (Corning, Tewksbury, MA) that had been sequentially coated with 150 µg/ml poly-d-lysine (Biomedical Technologies, Stoughton, MA) and 20 µg/ml laminin (Sigma, St. Louis, MI). For the non-adherent culture of neurospheres, CNS progenitors were plated at a density of 1.33–2.67 cells/µl (2000–4000 cells per 35 mm well) in ultra-low binding 6-well plates (Corning). Cells were initially cultured for 7 to 9 days in 'self-renewal medium' to promote the formation of undifferentiated colonies. This medium was a 5:3 mixture of DMEM-low:neurobasal medium, supplemented with 20 ng/ml recombinant human bFGF (R&D Systems, Minneapolis, MN), 20 ng/ml epidermal growth factor (EGF) (R&D Systems), 10% chick embryo extract (CEE; made as described [*Stemple and Anderson, 1992*]), 1% N2 supplement (GIBCO, Grand Island, NY), 2% B27 supplement (GIBCO), 50 mM 2-mercaptoethanol, and penicillin/

streptomycin (BioWhittaker, Walkersville, MD). After 7–9 days in self-renewal medium, cultures were fed with 'differentiation medium' and allowed to grow for an additional 4 to 6 days. Differentiation medium contained 10 ng/ml of bFGF (instead of 20 ng/ml), 5% fetal bovine serum (GIBCO), no EGF, and no CEE. After being grown in self-renewal medium, neurospheres were transferred to adherent cultures containing differentiation medium before being stained to assess multilineage differentiation. When *Lin28a Tg* neurospheres were cultured, 2 µg/ml doxycycline were added to the culture medium to sustain *Lin28* transgene expression. All cultures were maintained at 37°C in 6% $CO_2$ incubators.

To quantify self-renewal potential, individual CNS neurospheres were dissociated by trituration then replated at clonal density (1.33 cells/µl) in nonadherent secondary cultures. Secondary neurospheres were counted 7 to 9 days later, then transferred to adherent cultures containing differentiation medium to measure the percentage of secondary neurospheres that could undergo multilineage differentiation.

For viral infection (sometimes lentivirus and sometimes retrovirus, depending on the experiment), CNS progenitors were plated at a high density of 10 cells/µl and cultured adherently in self-renewal medium. After 48 hr, viral supernatant was added for 24 hr then switched to fresh self-renewal medium for a further 24 hr. Cells were harvested by incubating for 1.5 min at 37°C in trypsin/EDTA and transferred to nonadherent cultures to form neurospheres for 7 days.

To measure mRNA decay, neurospheres formed by cells dissected from E13.5 dorsal telencephalon were plated adherently and cultured for 2 days in self-renewal medium. Then fresh self-renewal medium with 10 µg/ml Actinomycin D was added and the cells were harvested at the indicated time points to examine mRNA levels by quantitative RT-PCR.

For explant cultures, pieces of E12.5 wild-type dorsolateral telencephalon were placed onto transwell plates (6.5 mm with 8.0 µm Pore Polycarbonate Membrane Insert, Corning) and cultured for 12 hr with 'explant culture medium' (a 5:3 mixture of DMEM-low:neurobasal medium, 1% N2 supplement, 2% B27 supplement, 50 mM 2-mercaptoethanol, and penicillin/streptomycin) supplemented with or without recombinant mouse Wnt-3a (100 ng/ml, R&D Systems), recombinant mouse Dkk-1 (200 ng/ml, R&D Systems), or recombinant mouse BMP-4 (100 ng/ml, R&D Systems).

## Generation of *let-7b* conditional mutant mice

To generate *let-7b* [flox/flox] mice, bacterial artificial chromosome (BAC) clones containing the *let-7b/c2* genomic locus were purchased (Invitrogen, Grand Island, NY) and a targeting vector was constructed using bacterial recombineering (*Copeland et al., 2001*; *Liu et al., 2003*). In this construct, the *let-7b/c2* genomic locus was flanked by loxP elements (see *Figure 7—figure supplement 2*). Neomycin resistance, diphtheria toxin fragment A (DT-A), and thymidine kinase cassettes were included for positive and negative selection. Bruce 4.G9 ES cells were electropolated with the targeting construct, positively selected with G418 (Gibco), and negatively selected with gancyclovir (cytovene from Syntex). Correctly targeted ES cells were identified by Southern blot and their karyotypes were assessed. Three independent euploid ES cell clones were injected into blastocysts obtained from B6(Cg)-*Tyr*[c-2j]/J mice (Jackson Laboratory, Bar Harbor, ME). The resulting male ES cell/mouse chimeras were bred with B6(Cg)-*Tyr*[c-2j]/J mice to obtain germline transmission. After germline transmission, the neo cassette was removed by crossing with B6.Cg-Tg(ACTFLPe)9205Dym/J mice (*Rodriguez et al., 2000*). Conditional *let-7b/c2* mutant mice (*Nestin-Cre*; *let-7b/c2*[flox/flox]) were obtained by breeding *Nestin-Cre* mice (*Tronche et al., 1999*) with *let-7b/c2* [flox/+] mice.

## Ribonucleoprotein immunoprecipitation (RIP) and RNA-Seq

CNS neurospheres formed by cells dissected from E13.5 *Imp1*[β-geo/β-geo] dorsal telencephalon were infected with pMIG-3XFLAG-Imp1 retrovirus. Immunoprecipitation of Imp1-FLAG fusion protein along with the mRNAs it bound was done using an RIP-Assay kit (MBL, Woburn, MA) according to the manufacturer's instructions. Briefly, neurospheres were washed with ice cold-PBS and lysed in buffer containing proteinase inhibitor cocktail (Sigma) and RNAse inhibitor (Roche, Madison, WI) on ice for 15 min. After clearing cell debris by centrifugation, supernatants were incubated with Protein G-Agarose for 60 min to absorb non-specific binding.

After eliminating Protein G-Agarose by centrifugation, small fractions of supernatants were saved as input, and the rest was incubated for 3 hr with anti-FLAG M2 Affinity Gel (Sigma). Immunoprecipitated fractions were washed with washing buffer and RNAs were eluted with solution I-IV, precipitated with ice-cold 2-propanol, washed with ice-cold ethanol, and suspend in RNAse free water according to the RIP-Assay kit manufacturer's instructions (MBL). cDNA synthesis and library construction were performed

using Ovation RNA-Seq System V2 and Encore NGS Library System I (Part Numbers 7102 and 300; NuGEN Technologies, Inc., San Carlos, CA) following the manufacturer's instructions. We used 100 ng total RNA from the FLAG immunoprecipitated fraction or input fraction from three independent experiments. Briefly, first strand cDNAs were synthesized by incubating RNAs with first strand reagents in the Ovation RNA-Seq System V2 kit by incubating at 65°C for 5 min, 4°C for 1 min, 25°C for 10 min, 42°C for 10 min, 70°C for 15 min. Second strand cDNAs were generated by incubating with second strand reagents in the Ovation RNA-Seq System V2 kit at 4°C for 1 min, 25°C for 10 min, 50°C for 30 min, 80°C for 20 min. Primers and nucleotides were eliminated from purified cDNAs by mixing with Agencourt RNAclean XP purification beads (Beckman coulter) followed by magnetic separation on a 96-well magnetic plate (ALPAQUA) for 5 min cDNA was amplified by incubating with SPIA reagents in the Ovation RNA-Seq System V2 kit at 4°C for 1 min, 47°C for 60 min, and 80°C for 20 min.

Amplified cDNAs were quantified with Quant-iT PicoGreen dsDNA reagent and kits (Molecular probes Inc., Eugene, OR), and sheared to 150–200 bp fragments using a Covaris S220 ultra-sonicator (Covaris, Woburn, MA) at 10% duty cycle, intensity set at 5, 100 cycles/burst for 5 min. Sheared cDNAs were then endo-repaired by incubating with endo-repair reagents in the Encore NGS Library System I kit at 25°C for 30 min, 70°C for 10 min. Primers and nucleotides were eliminated from cDNAs by mixing with Agencourt RNAclean XP purification beads (Beckman coulter, Brea, CA) followed by magnetic separation on a 96-well magnetic plate (ALPAQUA, Beverly, MA) for 5 min. Next adaptors were ligated by incubating with ligation adaptor reagents in the Encore NGS Library System I kit at 25°C for 10 min, then cDNAs were amplified by incubating with amplification reagents in the same kit at 72°C for 2 min, followed five cycles at 94°C for 30 s, 55°C for 30 s, 72°C for 1 min, 10 cycles at 94°C for 30 s, 63°C for 30 s, 72°C for 1 min, and 72°C for 5 min.

cDNA libraries were sequenced to 50-fold coverage using a Hiseq 2000 Sequencing System (Illumina, Inc., San Diego, CA). The raw sequence data were assessed for quality using FASTQC software (http://www.bioinformatics.bbsrc.ac.uk/projects/fastqc/) and aligned to the mouse reference genome MM9 (build 37) using TopHat (*Langmead et al., 2009*; *Trapnell et al., 2009*). Differences in RNA abundance were assessed using the CuffDiff algorithm in the CuffLinks software (*Roberts et al., 2009*; *Trapnell et al., 2010*). Parameter settings were: fragment-bias-correct (mm9.fa), compatible-hits-norm, multi-read-correct, and upper-quartile-norm. Using a locally derived perl script, we selected genes as being differentially expressed if they showed test status 'OK', FDR <= 0.05, and fold-change of at least 2.0.

## Isolation of CNS progenitors

E12.5-E18.5 dorsal telencephalons were removed and dissociated by incubating for 4 min at 37°C in 0.5 ml/ml DNAse1 (Sigma) in Ca, Mg-free HBSS, and resuspended in staining medium: L15 medium (Gibco) containing 1 mg/ml BSA (Sigma A-3912, St. Louis, MO), 10 mM HEPES (pH7.4) and penicillin/streptomycin (BioWhittaker, Walkersville, MD). After centrifuging (200×$g$ for 4 min), the cells were gently triturated, filtered through nylon screen (45 nm, Sefar America, Kansas City, MO), counted by hemocytometer, and plated.

## In utero knockdown

E12.5 or E14.5 timed-pregnant female C57BL/6 mice were anesthetized with isofluorane and their uteri were exposed. 0.5–1 µl of lentiviral supernatant, including 10 µg/ml Polybrene (Millipore) and 0.05% fast green (Sigma F-7258, St. Louis, MO), were delivered into the lateral ventricle of telencephalons of each embryos using glass capillaries. After injection, uteri were placed back into the abdomen and the wounds were closed with surgical sutures (Tevdek II 3-0, DEKNATEL, Gurnee, IL). Three days later, embryos were fixed and sectioned for immunostaining.

## Immunocytochemistry

CNS neurospheres were tested for multipotency by replating one neurosphere per well of 48-well plates coated with poly-d-lysine and laminin. The adherent neurospheres were allowed to differentiate for 4 to 6 days, then incubated first in anti-O4 antibody (1:800 ascites, Developmental Study Hybridoma Bank, Iowa city, IA), and then fixed in acid ethanol (5% glacial acetic acid in 100% ethanol) for 20 min at −20°C. After blocking and washing, the cultures were stained in donkey anti-mouse-IgM secondary antibody conjugated to horse radish peroxidase (Jackson Immunoresearch, West Grove, PA), followed by Nickel diaminobenzidine staining. Then cultures were stained with Tuj1 (1:500 Covance, Princeton NJ) and anti-GFAP (1:200, Sigma G-3893) primary antibodies followed by Alexa-Fluor 488 or 555 conjugated goat anti-mouse IgG$_1$ and goat anti-mouse IgG$_{2a}$ secondary antibodies (1:1000 each, Molecular Probes Inc., Eugene, OR).

For analyses of cell proliferation in culture, cells were pulsed with 10 µM BrdU (Sigma) for 20 min, fixed in 70% ethanol for 30 min at −20°C, and stained with an anti-BrdU antibody (1:200 Caltag, Burlingame, CA) overnight at 4°C. Alexa-Fluor 488 conjugated goat anti-mouse IgG2a secondary antibody (Molecular Probes; 1:1000) was then stained for 1 hr at room temperature. For caspase-3 staining, cultures were fixed for 10 min at room temperature in 4% paraformaldehyde, blocked, then stained with an anti-activated caspase 3 antibody (1:1000, Pharmingen, San Diego, CA) overnight at 4°C. Alexa-Fluor 555 conjugated goat anti-rabbit IgG secondary antibody (Molecular Probes; 1:1000) was then stained for 1 hr at room temperature. In all cases, cell were counter stained for 10 min at room temperature with 10 µg/ml DAPI (Sigma D-8417).

For X-gal staining of neurospheres, E12.5 or E18.5 CNS neurospheres were fixed with 1% paraformaldehyde plus 0.2% glutaraldehyde for 5 min at 4°C, and incubated for 1 hr at 37°C in staining solution: PBS containing 2 mM 5-bromo-4-chloro-3-indolyl-beta-D-galactosidase (X-dgal; Molecular Probes, Eugene OR, USA), 2 mM MgCl₂, 5 mM potassium ferrocyanide, 5 mM potassium ferricyanide, and 0.02% NP-40. In some cases, neurospheres were fixed with 4% paraformaldehyde for 10 min at 4°C, cryoprotected in 30% sucrose, embedded in OCT compound (Sakura Fineteck Inc., Torrance, CA) and frozen. Then 10 µm sections were cut and stained with chick anti-beta-galactosidase antibody (1:2000, BGL-1040, Aves Labs Inc., Tigard, OR) and anti-nestin antibody (1:400, MAB353, Millipore, Billerica, CA). Alexa-Fluor 488 conjugated goat anti-mouse IgG2a secondary antibody (Molecular Probes; 1:1000) was then stained for 1 hr at room temperature.

## Immunohistochemisty in tissue sections

Brains were fixed in 4% paraformaldehyde at 4°C overnight, cryoprotected in 30% sucrose, embedded in OCT compound, and frozen. 12 µm sections were cut, then pre-blocked for at least 1 hr at room temperature in blocking solution (PBS containing 5% goat serum, 0.2% bovine serum albumin, and 0.5% Triton X-100), incubated with primary antibody at 4°C overnight, followed by washing, and incubation in secondary antibody for 1 hr at room temperature. For some antigens (Ki67, BrdU), sections were boiled before blocking in 10 mM sodium citrate (pH 6.0) for 10 min to retrieve antigens. Sections were counter stained in 2.5 µg/ml DAPI for 10 min at room temperature, then mounted using ProLong antifade solution (Molecular Probes Inc., Eugene, OR). Primary antibodies included those against beta-galactosidase (1:2000), Tuj1 (1:1000), phospho-Histone H3 (Cell Signaling Technology Inc., Danvers, MA, 1:200), Pax6 (1:1000, Millipore, Billerica, MA), Tbr2 (1:200, Abcam, Cambridge, MA), GFAP (1:100,0, DAKO, Carpinteria, CA), Cyclin D1 (Thermo Scientific, Fremont, CA, 1:200), BrdU (1:200, Accurate Chemical, Westbury, NY), Ki67 (1:200, clone B56, BD Biosciences, San Jose, CA), and TAG-1 (1:400 ascites, Developmental Study Hybridoma Bank, University of Iowa, Iowa city, IA). For secondary antibodies, Alexa-Fluor 488 or 555 or 647 conjugated antibodies were used (1:1000 each, Molecular Probes Inc., Eugene, OR). TUNEL staining was performed using the Apoptag fluorescein In Situ Apoptosis Detection kit (Millipore).

For cell cycle exit analysis, E13.5 pregnant dams were injected intraperitoneally with 50 mg BrdU/kg body mass. 24 hr later, E14.5 pups were dissected and brains were fixed and processed as described above. For X-gal staining, E10.5-E12.5 mouse embryos or E14.5-P0 brains were fixed with 1% paraformaldehyde plus 0.2% glutaraldehyde for 15 min at 4°C. Then whole brains or cryosections were incubated in staining solution at 37°C for 4 to 16 hr as described above.

For in situ hybridization to *Imp1* and *Hmga2* transcripts in tissues, brains were fixed in 4% paraformaldehyde at 4°C overnight, cryoprotected in 30% sucrose, embedded in OCT, and frozen. 12 µm sections were cut, pretreated with 2 µg/ml Proteinase K at 37°C for 20 min, with 0.2N HCl for 10 min at room temperature, with 0.1M triethanolamine-HCl for 10 min at room temperature and Digoxigenin-labeled antisense probe at 55°C overnight. The next day, sections were washed with 2 × SSC for 30 min at 55°C, with 0.2 × SSC for 40 min at 55°C, blocked with 20% goat serum for 1 hr, and incubated with anti-Digoxigenin-labeled-AP (Alkaline phosphatase) Fab fragment (1:2000, Roche) for 60 min at room temperature. Sections were washed with Tris buffered saline (pH 9.5) with 0.1% Tween-20 for 30 min at room temperature, and incubated with 0.5 µl/ml NBT (nitro-blue tetrazolium chloride) plus 3.5 µl/ml BCIP (5-Bromo-4-Chloro-3′-Indolylphosphatase *p*-Toluidine salt) (Roche).

## Western blots and quantitative real-time PCR (qPCR)

Cells or tissues were resuspended in ice-cold cell lysis buffer (Cell Signaling Technology, Danvers, MA) with protease inhibitor cocktail (Sigma), and incubated for 20 min on ice. SDS PAGE was done in

4–20% Tris-Glycine Gels (Invitrogen) and transferred to PDVF membranes (Millipore). The membranes were blocked in Tris buffered saline with 0.05% Tween-20 and 5% milk powder, incubated with primary and secondary antibodies, and washed following standard procedures. Horse radish peroxidase conjugated secondary antibodies were detected by Supersignal West Femto Chemiluminescent Sustrate (Pierce). Primary antibodies were rabbit anti-IGF2BP/IMP1 (MBL, 1:2000), mouse anti-β-Catenin (1:2500, BD Biosciences), mouse anti-Synaptotagmin 1 (Abcam, 1: 2000), rabbit anti-LRRTM2 (1:2000), mouse anti-UNC5D (Abcam, Cambridge, MA, 1:1000), rat anti-OMgp (1:2500, R&D Systems), rabbit anti-HMGA2 (1:2000, a generous gift from M Narita and S Lowe), mouse anti-FLAG (Sigma, M2 1:5000), human anti-ribosomal P antigen (Immunovision, Springdale, AR, 1: 20000), and mouse anti-α-tubulin (1:10000, Sigma).

Quantitative RT-PCR was performed as described previously (*Nishino et al., 2008*). Primers used for amplification are listed in *Supplementary file 1B*. For *let-7b*, small RNAs (<200 nt) were extracted with mirVana miRNA isolation kit (Ambion, Grand Island, NY), and RT-PCR was performed with specific primers and probes supplied in Taqman MicroRNA Assay kits (Applied Biosystems, Grand Island, NY).

## Generation of virus

The *Imp1-GFP* vector was constructed by subcloning mouse *Imp1* cDNA (corresponding to NCBI NM_009951 from 312 to 7455, including the *Imp1* ORF and *let-7* binding sites in the 3′-UTR but lacking the polyadenylation signal) with N-terminal 3XFLAG into Bgl2-XhoI sites of the retroviral vector pMIG (MSCV-IRES-GFP). For *Imp1* (3′-UTR del)-*GFP* vector construction, the *Imp1* ORF (NCBI NM_009951 from 312 to 2045) was used instead. For *Imp1-β-geo* fusion protein+*GFP* vector construction, the 5′-fragment of *Imp1* ORF (NCBI NM_009951 from 312 to 547) and *β-geo* ORF were subcloned with N-terminal 3XFLAG into Bgl2-XhoI sites of retroviral vector pMIG. The *CyclinD1-GFP* vector was constructed by subcloning mouse *cyclin D1* ORF (NCBI NM_007631 from 233 to 1120) into the EcoRl site of the retroviral vector pMIG. Constructs for *let-7b* or *Hmga2* were described previously (*Nishino et al., 2008*). Viral supernatants were prepared by co-transfecting proviral plasmids and packaging vectors (pCL-Eco and pC1-VSVG) into 293T cells by standard calcium phosphate precipitation methods. The supernatant was collected after 72 hr and incubated with Retro-X Concentrator (Clontech, Mountain View, CA) at 4°C overnight. The next day, a viral pellet was obtained by centrifugation at 1,500×g in a Beckman JS 5.3 rotor for 45 min at 4°C and resuspended in a 5:3 mixture of DMEM-low:neurobasal medium for addition to culture medium.

## Chromatin Immunoprecipitation (ChIP)

CNS neurospheres were formed by non-adherently culturing E12.5 wild-type dorsal telencephalon cells. The neurospheres were then plated adherently on 100 mm dishes coated with poly-d-lysine and laminin, and cultured in self-renewal medium (see above for composition) for additional 2 days. ChIP was done using the EZ ChIP Chromatin Immunoprecipitation kit (Upstate, Billeria, CA) according to the manufacturer's instructions. Briefly, cells were fixed with 1% paraformamide for 10 min at room temperature, washed with ice cold PBS, and lysed in SDS lysis buffer supplemented with protease inhibitor cocktail. Then DNA was sheared using a sonicator (Virsonic 100, VirTis Inc., Warminster, PA), and incubated with Protein G-Agarose for 60 min at 4°C to absorb non-specific binding. After eliminating Protein G-Agarose by centrifugation, small fractions of supernatants were saved as input, and the rest were incubated overnight at 4°C with either monoclonal anti-TCF4 antibody (clone 6H5-3, Millipore) or normal mouse IgG (supplied in EZ ChIP kit). The next day, immunoprecipitated fractions were collected by incubating with Protein G-Agarose for 60 min at 4°C, washed with washing buffer, eluted with elution buffer, and reverse-crosslinked in 0.2 M NaCl at 65°C overnight. Then DNA was purified using an affinity column, and subjected to PCR. PCR primers are listed in *Supplementary file 1B*.

## Luciferase assay

An *Imp1* genomic DNA fragment (corresponding to −1980 to +1 base pairs in *Figure 5—figure supplement 1*) was subcloned into pBluesript SK(+) (Stratagene, La Jolla, CA), then site A (from −1906 to −1901 bp), site B (from −487 to −481 bp), or both site A and site B were eliminated by PCR. Intact or mutated fragments were subcloned into pGL3-Vector (Promega, Madison, WI) to generate luciferase reporter plasmids. These reporter plasmids, TOPflash (Millipore), or empty pGL3-Vector were mixed with pRL-TK vector (Promega) at a 10:1 ratio, and transfected to P19 embryonic carcinoma cells (ATCC) using lipofectamine 2000 (Invirogen). After transfection for 36 hr,

cells were exposed to medium supplemented with 20 mM LiCl and cultured for an additional 12 hr. A dual luciferase assay was conducted using Dual-Luciferase Reporter Assay System (Promega) according to the manufacturer's instructions. Briefly, cells were washed with PBS, and lysed in 1X passive lysis buffer at ambient temprature for 20 min. Cell lysates were mixed with Luciferase Assay Reagent II and firefly luciferase activity was measured by microplate reader (FLUOstar Omega, BMG LABTECH, Cary, NC). Next, 1X Stop & Glo Reagent was added and mixed, and *Renilla* luciferase activity was measured.

## Polysome analysis

CNS neurospheres were formed by non-adherently culturing E13.5 wild-type dorsal telencephalon cells then plated adherently on 100 mm dishes coated with poly-d-lysine and laminin, and cultured in self-renewal medium for an additional 2 days. Cells were treated with 0.1 μg/ml cycloheximide (Sigma) for 3 min, washed three times with PBS plus 0.1 μg/ml cycloheximide, and then lysed on ice for 15 min in polysome extraction solution (10 mM Tris [7.4], 15 mM $MgCl_2$ ,0.3M NaCl, 1% Triton X-100, and 10 μg/ml heparin [Sigma]) plus 0.1 μg/ml cycloheximide. After clearing cell debris by centrifugation, small aliquots of lysates were saved as input controls and the rest were loaded on top of a 10–50% sucrose gradient in polysome extraction buffer plus 0.1 μg/ml cycloheximide, and spun down at 4°C at 35,000 rpm for 190 min in an SW41 rotor (Beckman Coulter). After ultracentrifugation, fractions were taken serially from the top (10% sucrose) to the bottom (50% sucrose). Each fraction was divided into half and either saved in Trizol (Invitrogen) for RNA extraction or subjected to methanol/chloroform protein extraction. To assess the shift of ribosomal proteins, parallel cultures were processed in solutions that contain 30 mM EDTA instead of cycloheximide.

## Genotyping of mice

Genotyping was performed by PCR following the manufacturer's instructions using Go Taq Flexi DNA polymerase (Promega) for *Imp1*, *Lin28a*, *let-7*, *Hmga2*, and *let-7b/c2* mutant mice and Choice-Taq DNA Polymerase (Denville Scientific Inc. Metuchen, NJ) for *Apc*, *β-catenin*, and *Nestin-Cre* mice. The PCR conditions for *Imp1*, *Hmga2*, and *let-7b/c2* mutant mice were 94°C for 2 min, then 33 cycles of 94°C for 30 s followed by 60°C for 30 s and 72°C for 1 min, with 72°C for 2 min at the end. For *Lin28a* and *let-7* inducible transgenic mice PCR genotyping conditions were 94°C for 2 min, then 35 cycles of 94°C for 30 s followed by 55°C for 30 s and 72°C for 1 min, with 72°C for 2 min at the end. For *Apc* mutant mice, 94°C for 3 min, then 34 cycles of 94°C for 1 min followed by 55°C for 1 min and 72°C for 2 min, with 72°C for 10 min at the end. For *β-catenin* mutant mice, 94°C for 3 min, then 34 cycles of 94°C for 1 min followed by 59°C for 1 min and 72°C for 2 min, with 72°C for 10 min at the end. For *Nestin-cre*, 94°C for 3 min, then 34 cycles of 94°C for 1 min followed by 63°C for 1 min and 72°C for 2 min, with 72°C for 10 min at the end. Primers used for genotyping are listed in *Supplementary file 1B*.

## Acknowledgements

This work was supported by the Cancer Prevention and Research Institute of Texas, the National Institute on Aging (R37 AG024945), the National Institutes of Neurological Disease and Stroke (NS40750), and the Howard Hughes Medical Institute. JN was supported by a postdoctoral fellowship from the Japan Society for the Promotion of Science. Thanks to Carla Green, Elain Shar, and Shihoko Kojima (University of Texas Southwestern Medical Center) for helping with the polysome analysis. Thanks to Brendan Tarrier (University of Michigan DNA Sequencing Core) for conducting NGS seqencing. Thanks to Richard McEachin (University of Michigan Center for Computational Medicine and Bioinformatics) and Zhiyu Zhao (University of Texas Southwestern Medical Center) for analysing NGS sequence data. Thanks to George Daley (Boston Children's Hospital) for providing *Lin28a* transgenic and *ilet-7* transgenic mice. Thanks to Rhonda Bassel-Duby (University of Texas Southwestern Medical Center) for providing pGL3 and pRL-TK vectors.

## Additional information

### Competing interests

SJM: Reviewing editor, *eLife*. The other authors declare that no competing interests exist.

## Funding

| Funder | Grant reference number | Author |
| --- | --- | --- |
| National Institute on Aging | R37 AG024945 | Sean J Morrison |
| Howard Hughes Medical Institute | | Sean J Morrison |
| National Institute of Neurological Disease and Stroke | NS40750 | Sean J Morrison |

The funders had no role in study design, data collection and interpretation, or the decision to submit the work for publication.

## Author contributions

JN, Designed and conducted all experiments and wrote the manuscript; SK, YZ, Maintained and genotyped APCflox, β-catenin flox, and human-GFAP Cre mice; HZ, Contributed to the experiments performed with inducible let-7 and Lin28a transgenic mice; SJM, Participated in the design and interpretation of all experiments and wrote the manuscript with JN

## Ethics

Animal experimentation: All mice used in this study were housed in the Unit for Laboratory Animal Medicine at the University of Michigan (UM) or in the Animal Resource Center at the University of Texas Southwestern Medical Center (UTSW). All procedures were approved by the UM Committee on the Use and Care of Animals (#8373) and by the UTSW Institutional Animal Care and Use Committee (2011-0104).

## Additional files

### Supplementary files

• Supplementary file 1. (**A**) mRNAs that were significantly enriched upon immunoprecipitation of IMP1. E13.5 $Imp1^{\beta\text{-geo}/\beta\text{-geo}}$ dorsal telencephalon cells were infected with pMIG-3XFLAG-Imp1 retrovirus, then expanded in culture, then transcripts bound to 3XFLAG-IMP1 protein were immunoprecipitated using anti-FLAG antibody. Shown are gene symbol, gene name, average transcript levels as a ratio of FLAG pulled-down fraction/input fraction, and p value. Data were collected from three replicates. We do not know how many of these mRNAs are actually regulated by IMP1 binding. (**B**) Primer sequences used in this study.

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
