## [Decision Letter]

Thank you for sending your work entitled “A network of heterochronic genes including Imp1 regulates temporal changes in stem cell properties” for consideration at *eLife*. Your article has been favorably evaluated by a Senior editor and 3 reviewers, one of whom, Leanne Jones, has agreed to reveal her identity.

The Senior editor, Janet Rossant, and the three reviewers discussed their comments before we reached this decision, and the Senior editor has assembled the following comments to help you prepare a revised submission.

Overall, the reviewers agreed that you have developed a strong set of data implicating the let7 target gene, *Imp1*, in neural stem cell proliferations during fetal brain development. However, there was a major concern with your conclusion that *Imp1* was cell-autonomously required for NSC self-renewal, both in vitro and in vivo. You used the gene trap *Imp1-βgeo* knockout mouse, which shows overall growth defects throughout the embryo (and the placenta) by E17.5. Given that *Imp1βgeo/βgeo* neurospheres only showed self-renewal defects in vitro when isolated at E18.5 and not E12.5, this raises the possibility that the effects could be secondary. E18.5 is a time-point where the severe growth defects could have already affected the NSCs non-cell autonomously before isolation, perhaps due to a systemic reduction in IGF2 (21). Given that WT *Imp1* expression is highest at E12.5, but nearly extinguished in the WT brain by E17.5, one might have expected that defects would have been more likely in the early rather than the late neurospheres. The fact that you only observed NSC defects at E18.5 in *Imp1βgeo/βgeo* mice, in vitro and in vivo throughout the manuscript, leaves open the possibility that the effects could be completely explained by non-cell autonomous effects. In order to prove cell-autonomy, neural-specific deletion of a floxed *Imp1* allele would be required. If you can undertake these experiments, this would strengthen your conclusions. If this experiment does not work, you would have to modify your claim of cell-autonomy. We understand that these experiments would take longer than two months unless you have already initiated the crosses.

---

## [Author Response]

*Overall, the reviewers agreed that you have developed a strong set of data implicating the let7 target gene,* Imp1*, in neural stem cell proliferations during fetal brain development. However, there was a major concern with your conclusion that* Imp1 *was cell-autonomously required for NSC self-renewal, both* in vitro *and* in vivo*. You used the gene trap* Imp1-βgeo *knockout mouse, which shows overall growth defects throughout the embryo (and the placenta) by E17.5. Given that* Imp1βgeo/βgeo *neurospheres only showed self-renewal defects* in vitro *when isolated at E18.5 and not E12.5, this raises the possibility that the effects could be secondary. E18.5 is a time-point where the severe growth defects could have already affected the NSCs non-cell autonomously before isolation, perhaps due to a systemic reduction in IGF2 (*[21]*). Given that WT* Imp1 *expression is highest at E12.5, but nearly extinguished in the WT brain by E17.5, one might have expected that defects would have been more likely in the early rather than the late neurospheres. The fact that you only observed NSC defects at E18.5 in* Imp1βgeo/βgeo *mice,* in vitro *and* in vivo *throughout the manuscript, leaves open the possibility that the effects could be completely explained by non-cell autonomous effects. In order to prove cell-autonomy, neural-specific deletion of a floxed* Imp1 *allele would be required. If you can undertake these experiments, this would strengthen your conclusions. If this experiment does not work, you would have to modify your claim of cell-autonomy. We understand that these experiments would take longer than two months unless you have already initiated the crosses*.

There are several issues to address here. The first important point is that although Figure 2 showed that neurospheres had self-renewal defects at E18.5 and not at E12.5, this should not be taken to mean that neurospheres did not exhibit defects prior to E18.5. We simply had not tested cultured neurospheres at intermediate time points because we focused our analyses on more definitive in vivo experiments at the intermediate time points. To address this we performed two experiments in which we examined the frequency and self-renewal of multipotent neurospheres cultured from E15.5 *Imp1* deficient and control telencephalon. We observed declines in the frequency and self-renewal of *Imp1* deficient neurospheres in both experiments. We have added these new data to Figure 2.

The second important point is that *Imp1* does continue to be expressed within stem cells from the dorsomedial telencephalon through E18.5. Its expression is extinguished neonatally. The manuscript shows IMP1 expression in the VZ of the dorsal telencephalon at E14.5, E16.5 (Figure 1—figure supplement 1) and in neurospheres cultured from the E18.5 dorsomedial telencephalon (Figure 7). As requested, we have also summarized the expression pattern and the anatomy schematically in Figure 1—figure supplement 2.

The third important point is that it is not accurate that we only observed NSC defects at E18.5 in the *Imp1βgeo/βgeo* mice. A number of figures in our manuscript show that we observed significant neural stem cell defects in vivo beginning at E12.5. These defects were associated with significant changes in the numbers of Tbr2+ progenitors beginning at E12.5 (Figure 3), the numbers of TuJ1+ neurons beginning at E12.5 (Figure 4), the number of Pax6+ stem cells beginning at E14.5 (Figure 3), the frequency of dividing cells beginning at E16.5 (Figure 2), and the cortical surface length beginning at E16.5 (Figure 2). The in vivo data demonstrate evidence of premature stem cell differentiation in vivo beginning at E12.5 and continuing throughout fetal development. There is no disconnect between the timing of *Imp1* expression in these cells and the timing with which premature differentiation is observed in vivo.

We believe that certain phenotypes were not observed until E16.5 or E18.5 because loss of IMP1 function causes neural stem cells to prematurely differentiate first into Tbr2+ intermediate progenitors (first detected at E12.5, Figure 3) and then into neurons (first detected at E12.5, Figure 4) and glia (first detected at E18.5, Figure 4). Since Tbr2+ intermediate progenitors are themselves dividing, this delays the onset of defects in proliferation and cortical growth, which were not observed until E16.5.

Unfortunately, nobody has ever made a floxed allele of *Imp1*, making it impossible for us to conditionally delete in neural stem cells. To address this we have instead injected virus bearing an shRNA against *Imp1* into the forebrain of fetal mice developing in utero. We confirmed that this shRNA profoundly reduced *Imp1* expression in cultured neurospheres (Figure 5). By transilluminating the uterus we were able to inject virus into the E12.5–E14.5 telencephalic ventricle, infecting a small minority of stem cells in the dorsal telencephalon. At E15.5–E17.5, we compared Ki67, Pax6, Tbr2, and Tuj1 staining by cells infected (GFP+) with the *Imp1* shRNA versus a control scrambled shRNA. Cells infected with the *Imp1* shRNA were significantly less likely to express Pax6 or Ki67 and significantly more likely to express Tbr2 or Tuj1 (Figure 5; Figure 5—figure supplement 1). This demonstrates that IMP1 acts cell autonomously within stem cells in the dorsal telencephalon to promote the maintenance of an undifferentiated state.

We agree that IMP1 is likely to have a mix of cell-autonomous and non-cell-autonomous effects on the growth of other tissues. We have acknowledged this in our manuscript; however, all of our data are consistent with cell-autonomous effects in the dorsal telencephalon.